# MME-VideoOCR: Evaluating OCR-Based Capabilities of Multimodal LLMs in Video Scenarios

**Yang Shi**[1,2◇*] **Huanqian Wang**[3◇] **Wulin Xie**[4◇] **Huanyao Zhang**[2◇] **Lijie Zhao**[5◇]
**Yi-Fan Zhang**[4◇‡] **Xinfeng Li**[6] **Chaoyou Fu**[7] **Zhuoer Wen**[2] **Wenting Liu**[2]
**Zhuoran Zhang**[2] **Xinlong Chen**[4] **Bohan Zeng**[2] **Sihan Yang**[8] **Yushuo Guan**[1]
**Zhang Zhang**[4] **Liang Wang**[4] **Haoxuan Li**[2] **Zhouchen Lin**[2]
**Yuanxing Zhang**[1‡] **Pengfei Wan**[1] **Haotian Wang**[3‡] **Wenjing Yang**[♠]

[1]Kling Team  [2]PKU  [3]THU  [4]CASIA  [5]CUHKSZ  [6]NTU  [7]NJU  [8]XJTU

◇ Core Contributor    ♠ Project Lead    ‡ Corresponding Author

https://mme-videoocr.github.io/

## Abstract

Multimodal Large Language Models (MLLMs) have achieved considerable accuracy in Optical Character Recognition (OCR) from static images. However, their efficacy in video OCR is significantly diminished due to factors such as motion blur, temporal variations, and visual effects inherent in video content. To provide clearer guidance for training practical MLLMs, we introduce **MME-VideoOCR** benchmark, which encompasses a comprehensive range of video OCR application scenarios. MME-VideoOCR features 10 task categories comprising 25 individual tasks and spans 44 diverse scenarios. These tasks extend beyond text recognition to incorporate deeper comprehension and reasoning of textual content within videos. The benchmark consists of $1,464$ videos with varying resolutions, aspect ratios, and durations, along with $2,000$ meticulously curated, manually annotated question-answer pairs. We evaluate 18 state-of-the-art MLLMs on MME-VideoOCR, revealing that even the best-performing model (Gemini-2.5 Pro) achieves only an accuracy of $73.7\%$. Fine-grained analysis indicates that while existing MLLMs demonstrate strong performance on tasks where relevant texts are contained within a single or few frames, they exhibit limited capability in effectively handling tasks that demand holistic video comprehension. These limitations are especially evident in scenarios that require spatio-temporal reasoning, cross-frame information integration, or resistance to language prior bias. Our findings also highlight the importance of high-resolution visual input and sufficient temporal coverage for reliable OCR in dynamic video scenarios.

## 1 Introduction

In recent years, the rapid advancement of Multimodal Large Language Models (MLLMs)[1–6] has garnered significant attention. These models, capable of processing and integrating information across various modalities (e.g., text, images, and video), have demonstrated considerable potential and significant value across a wide range of real-world applications[7–14].

Optical Character Recognition (OCR) [15], a fundamental technology in visual understanding, serves as a crucial link for enabling structured comprehension of image and video content. It transforms visual information into computationally analyzable semantic data. Within cross-modal learning, OCR provides critical feature support for text-visual alignment, directly impacting the performance of

---

*Work done during an internship at Kling Team.

39th Conference on Neural Information Processing Systems (NeurIPS 2025) Track on Datasets and Benchmarks.

Figure 1: **An example in MME-VideoOCR**. The task requires the MLLM to first recognize the textual information distributed across multiple video frames, and then to perform semantic understanding and reasoning over the extracted text to accurately determine the correct answer. The correct information is marked in blue, while misleading information is marked in red.

Table 1: **Key differences between MME-VideoOCR and prior video-based OCR benchmarks**. MME-VideoOCR features a larger number of task types and scenarios, employs fully manual annotations to ensure reliability, supports bilingual content for broader coverage, and enables comprehensive evaluation across perception, understanding, and reasoning.

| Benchmarks | #Videos | #QA | #Tasks | #Scenarios | Annotation | Bilingual | Perception | Understanding | Reasoning |
|---|---|---|---|---|---|---|---|---|---|
| OCR Benchmark [23] | 25 | 1,477 | 1 | 20+ | M | ✗ | ✓ | ✗ | ✗ |
| FG Bench [24] | 1,028 | 2,961 | 6 | 20+ | A&M | ✗ | ✓ | ✓ | ✗ |
| **MME-VideoOCR** | 1464 | 2,000 | 25 | 44 | M | ✓ | ✓ | ✓ | ✓ |

downstream tasks [16, 17]. Previous OCR-based benchmarks [18–22] primarily focus on evaluating the OCR-based capabilities of MLLMs in static image scenarios. Several studies [23, 24] have initiated preliminary investigations into video scenarios. However, they typically concentrate on perceiving textual content, often neglecting text-based understanding and reasoning.

Considering the unique challenges of video understanding tasks, a comprehensive video OCR evaluation must address three key issues, as illustrated in Figure 1. Firstly, textual information in videos can appear in various forms—such as foreground text, background scenery, on-screen annotations, watermarks, and floating overlays. This requires models to establish robust spatio-temporal visual-text associations and to effectively identify and extract relevant textual information from these diverse and often noisy sources across different shots. Secondly, critical textual information in videos is often distributed across multiple frames, rather than appearing in a single static image. Therefore, models must be capable of effectively recognizing, integrating, and understanding text content over time, leveraging temporal context to reconstruct and interpret fragmented or sequentially presented information. Thirdly, as task complexity increases, models must be able to reason over the recognized text. This reasoning ability is essential for deeper video understanding and remains a significant challenge for current MLLMs.

In this paper, we propose the MME-VideoOCR benchmark, which provides a comprehensive evaluation framework for OCR tasks in video scenarios. Recognizing the limitations of current OCR tasks in existing evaluation datasets, MME-VideoOCR encompasses 10 task categories and 25 specific tasks, incorporating a substantial number of actively collected or custom-created videos. As shown in Table 1, MME-VideoOCR consists of 1,464 videos, paired with 2,000 diverse and accurately human-annotated question-answer (QA) pairs. The tasks require answers based on both localized key information and a holistic understanding of the entire video.

The main contributions are summarized as follows:

1. MME-VideoOCR introduces a diverse set of video OCR tasks, utilizing manually quality-controlled videos and question-answer pairs. These tasks span multiple dimensions, such as perceptual accuracy, contextual comprehension, and cross-frame reasoning, which together enable a comprehensive evaluation of MLLMs' OCR capabilities in video scenarios.

2. We evaluate 18 state-of-the-art MLLMs, including publicly available models ranging from 7B to 78B in size, as well as closed-source models like GPT-4o and Gemini-2.5 Pro. The results demonstrate strong discriminative power and the challenges posed by MME-VideoOCR. Regarding discriminative power, the worst-performing model, LLaVA-OneVision 7B, has an accuracy of $46.0\%$, while the best-performing model achieves an accuracy of $73.7\%$, showing a significant gap in performance. Regarding task difficulty, on several tasks we designed, such as Cross-Frame Text Understanding and Text-Based Video Understanding, most models score below $60\%$.

3. The evaluation results further reveal significant deficiencies in current models on OCR tasks that require spatio-temporal reasoning and cross-frame information integration, thereby indicating a critical direction for MLLM optimization. Moreover, both high-resolution visual inputs and sufficient temporal coverage are essential for achieving reliable OCR performance in dynamic video settings. Notably, MLLMs exhibit a strong language prior bias during text recognition, frequently favoring semantically plausible outputs over visually accurate transcriptions.

## 2 Related Work

### 2.1 Multimodal Large Language Models

Through the integration of a vision encoder into Large Language Models (LLMs) and pretraining on large-scale multimodal data, MLLMs exhibit strong OCR capabilities, making them well-suited for downstream tasks such as document understanding [25, 26], key information extraction [17], and scene text recognition [27, 28]. Building on this foundation, some recent MLLMs [6, 29–33] have further extended their capabilities to handle video inputs, enabling them to process dynamic visual information. This advancement enables MLLMs not only to recognize text in static images, but also to extract text-related information from videos and leverage it for more effective video understanding. However, the increased visual complexity, dynamic content, and temporal dependencies inherent in video characteristics impose greater demands and challenges on MLLMs [3, 34]. Therefore, a comprehensive and effective evaluation of their OCR-based capabilities in video scenarios is crucial.

### 2.2 OCR Benchmarks for Multimodal Large Language Models

Most existing benchmarks [35] are designed to evaluate the OCR capabilities of MLLMs in static image scenarios, including TextVQA [18], OCR-VQA [19], SEED-Bench2-Plus [20], OCRBench [21] and OCRBench v2 [22]. A few works [23, 24] have extended to video, but they cover only a narrow range of task types, lack diversity in video content, and provide limited insight into the unique characteristics of OCR-based tasks in video scenarios. Moreover, these benchmarks emphasize text recognition while overlooking text-based understanding and reasoning. Some scene-text video QA benchmarks [27, 36, 37] incorporate textual cues into visual QA. However, they often overlook fine-grained text perception, including temporal grounding and attribute recognition, and do not fully evaluate the potential of text as a central driver for video understanding. Moreover, as they focus solely on scene-text understanding, which represents only a narrow application scenario, this is far from sufficient for a comprehensive evaluation of MLLMs' OCR-based capabilities. These benchmarks are also limited in video diversity, task variety, and their exploration of the unique characteristics of OCR-based tasks in video scenarios.

## 3 MME-VideoOCR

### 3.1 Task Definition

Challenges inherent in video data, such as motion blur, inter-frame interference, the complexity of cross-modal alignment, difficulties in tracking content across shots, and limited generalization in noisy scenes. These issues pose significant obstacles for both video coding [38, 39] and semantic perception [40, 41], critically impacting the accuracy of MLLMs. To rigorously assess and foster

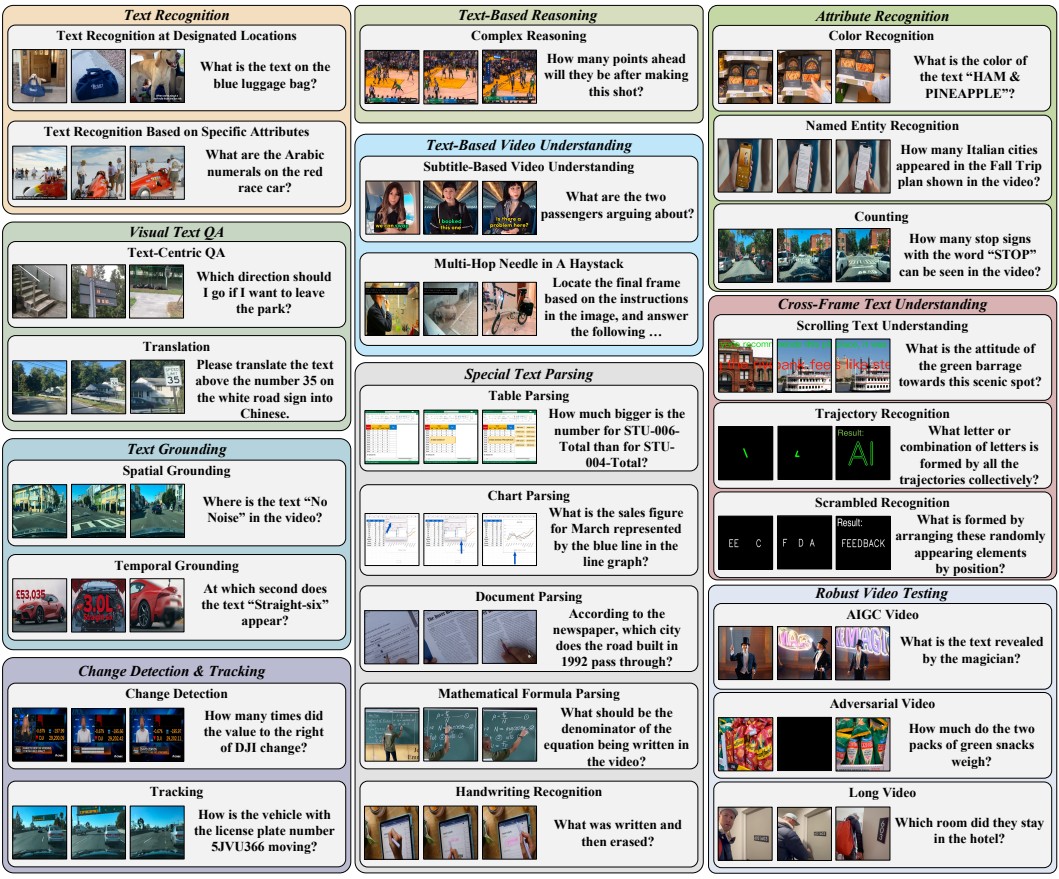

Figure 2: **Example videos and their annotated questions from the MME-VideoOCR benchmark**, encompassing 25 tasks across 10 categories. Each task is designed to evaluate models' capabilities in various aspects such as text recognition, localization, reasoning, and comprehensive video understanding. The figure displays representative video samples and their corresponding questions.

advancements in MLLMs against these challenges, we introduce MME-VideoOCR, a comprehensive benchmark comprising 25 distinct tasks across 10 categories (details can be found in Appendix B.1). Figure 2 showcases representative examples, illustrating the specific nature and scope of each task.

**Text Recognition** involves *Text Recognition at Designated Locations* and *Text Recognition Based on Specific Attributes* to evaluate the fine-grained text recognition capability.

**Visual Text QA** employs *Text-Centric QA* and *Translation*. Both tasks challenge the model's ability to not only perceive but also comprehend multimodal semantics.

**Text Grounding** introduces *Spatial Grounding* and *Temporal Grounding* to assess the model's ability on localizing and interpreting text across both spatial-temporal dimensions within dynamic scenes.

**Attribute Recognition** is composed of three tasks: *Color Recognition*, where models are expected to identify the color of the text; *Named Entity Recognition*, which focuses on extracting and classifying named entities; and *Counting*, where models must accurately identify the number of textual elements that meet specified criteria.

**Change Detection & Tracking** contains *Change Detection* and *Tracking* to identify textual changes over time and monitor text elements as they change position across frames, respectively.

**Special Text Parsing** includes five tasks: *Table Parsing*, *Chart Parsing*, *Document Parsing*, *Mathematical Formula Parsing*, and *Handwriting Recognition*. These tasks require models to accurately identify and understand text with either special structures or highly variable visual forms.

**Cross-Frame Text Understanding** includes three subtasks: *Scrolling Text Understanding*, which focuses on recognizing dynamic text streams that move across frames and may only be fully readable when aggregated over time; *Trajectory Recognition*, where the motion path of an object in the

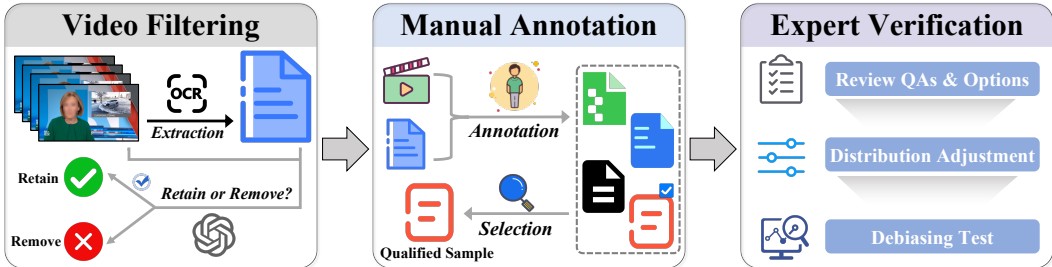

Figure 3: **Overview of the MME-VideoOCR construction process**. Video filtering ensures sufficient visual dynamics and meaningful textual content. Manual annotation provides high-quality QA pairs, and expert verification further enhances sample reliability and mitigates potential biases.

video forms a recognizable text, and the model must interpret this trajectory as the intended message; *Scrambled Recognition*, which involves identifying and reconstructing a complete text from characters that appear out of order across different positions in the video frames.

**Text-Based Reasoning** requires models to go beyond surface-level understanding by synthesizing dispersed cues, identifying implicit relation, and resolving ambiguity or misleading content.

**Text-Based Video Understanding** introduce a) *Subtitle-Based Video Understanding* which resembles real-world scenarios like conversations, tutorials, or news, where subtitles provide key information that visuals alone cannot capture; b) *Multi-Hop Needle in A Haystack* which requires reasoning over multiple pieces of subtitle content to find the correct answer.

**Robust Video Testing** contains three specialized video types: *AIGC Videos*, *Long Videos*, and *Adversarial Videos*. *AIGC Videos*, generated by AI systems [42], assess model adaptability to increasingly common synthetic content. *Long Videos* test the ability to extract relevant information from lengthy sequences with substantial redundancy. *Adversarial Videos* strategically insert all-black frames into normal videos, designed to mislead the MLLMs.

## 3.2 Benchmark Construction

**Video Collection & Filtering**. MME-VideoOCR covers as many diverse scenarios as possible in order to provide a comprehensive evaluation. To achieve this, we employ three distinct data collection methods, balancing diversity and efficiency in the construction of the benchmark.

*Reconstructing from Existing Video QA Data.* To maximize data collection efficiency, we leverage existing text-based video QA datasets, including BOVText [43], M4-ViteVQA [36], NewsVideoQA [44], LSVTD [45], RoadText-1K [46], RoadTextVQA [37], EgoTextVQA [27], NIAH-Video [47], and DSText [48]. For each video in these datasets, we uniformly sample 5-10 frames and extract the text using PaddleOCR [49]. The sampled frames, along with the extracted text, are then processed using GPT-4o to evaluate whether the video exhibits sufficient visual dynamics and contains semantically meaningful text. Only videos that meet these criteria are retained for further use.

*Manual Collection of Publicly Available Videos*. Existing benchmarks often lack the diversity needed to fully satisfy the requirements of our 25 OCR tasks. Therefore, we manually collect additional data from publicly available online sources (e.g., YouTube, Bilibili, Kuaishou) to further enhance diversity and ensure coverage of specific scenarios that are underrepresented in current datasets, such as webpages, charts, and mathematical formula derivations. Additionally, since most existing MLLMs are primarily trained on horizontally oriented videos, we intentionally include vertically formatted video content to improve distributional balance and better reflect real-world usage scenarios.

*AI-Generated Videos*. The task of recognizing and understanding the text in AI-generated videos is becoming more critical. To cover this emerging scenario, we manually create a set of videos designed to diversify the dataset and introduce controlled challenges. We initially generated $2,000$ everyday phrases. These phrases were then expanded into scene descriptions using Llama3.1-8B [50], with the requirement that each scene must incorporate the corresponding text and include a narrative element detailing its appearance or disappearance. Subsequently, these descriptions were provided to Wan [42] for text-to-video generation. From the resultant videos, we selected those exhibiting accurate text rendering, high visual-scene integration, and plausible narratives for our evaluation set. These videos are not only useful for evaluating the model's ability to understand AI-generated

content but also address specific cases that are difficult to obtain from existing datasets or online sources, such as occluded text revealed over time.

**Manual Annotation**. In order to circumvent errors and biases that may arise from model-based annotations [51, 52], we opt for manual annotation to ensure the dependability of our samples. Human annotators are tasked with carefully examining each video and developing 3-4 QA pairs per video, adhering to the specified task requirements. Next, a second expert implements a selection process, retaining 1-2 high-quality QA pairs per video. This sequential two-stage screening process is expected to substantially ensure the generation of high-quality QA pairs exhibiting significant relevance, clarity, and challenging attributes, effectively preventing biases from individual annotators.

**Expert Verification**. To uphold the highest data quality, expert annotators meticulously verify the constructed dataset against stringent standards. This verification process specifically addresses potential issues such as *ambiguous questions*, *inaccurate answers*, and *insufficiently challenging problems*. Initially, annotators review and rectify any errors or ambiguities within the QA pairs. Subsequently, for multiple-choice questions, they thoroughly assess all options, confirming that each is meaningful, poses an appropriate level of challenge, and functions as a plausible distractor. To mitigate potential biases stemming from imbalanced answer option frequencies [53, 54], we ensure a uniform distribution of correct answers across all options. Furthermore, to identify and eliminate residual biases that could compromise evaluation reliability, we conduct a dedicated debiasing test, as detailed in Section 3.4. The complete data construction process is illustrated in Figure 3.

## 3.3 Evaluation Criteria

Considering the characteristics of different tasks, we employ three distinct evaluation metrics to balance accuracy and efficiency in evaluation.

**Containment Match**. For *Text Recognition* and *Handwriting Recognition*, where the model must accurately identify the recognized text, we simply check whether the ground truth appears in model's response. This straightforward yet effective method is widely adopted in previous work [21, 24, 55].

**GPT-Assisted Scoring**. In the *Translation* task, multiple valid answers may exist. These answers may vary in form but remain consistent in meaning. To ensure flexibility and prevent unnecessary constraints on the model, we incorporate GPT-Assisted Scoring. Given the reference answer and the model's response, GPT-4o-0806 [56] serves as the evaluator, assessing their consistency and assigning a binary score of either $0$ or $1$. The prompt is shown in Appendix B.3.

**Multiple-Choice**. Tasks like *Visual Text QA* and *Spatial Grounding* allow for highly flexible responses. Since both Containment Match and GPT-Assisted Scoring may introduce evaluation errors, we use a multiple-choice format for assessment. In this setting, the model only needs to select the most appropriate option, which simplifies evaluation and reduces ambiguity in scoring. We use a common prompt as shown in Appendix B.3.

## 3.4 Debiasing Test

Since the underlying LLM of an MLLM is pretrained on large-scale textual corpora, it may introduce biases into the evaluation [53]. One source of bias arises from textual priors. For example, when asked "`What does the text on the red warning sign say?`", the model is more likely to answer "``STOP``" than "`EXIT`" due to common co-occurrence patterns in pretraining data. Another issue is potential knowledge leakage. For instance, for a question like "`According to the video, when was the United States founded?`", the model may provide the correct answer without relying on visual input, thereby compromising the reliability of the evaluation.

To mitigate these biases, we introduce a debiasing test designed to quantify and minimize the influence of textual priors and knowledge leakage. Specifically, we evaluate Qwen2.5-VL-7B [30] by presenting questions and options without providing any meaningful visual input. Under this setup, the model relying solely on textual priors should ideally achieve accuracy close to $0\%$ for tasks evaluated via Containment Match and GPT-Assisted Scoring, indicating minimal textual bias. For multiple-choice questions, random guessing should yield an accuracy close to $25\%$. After each debiasing test iteration, expert annotators review and revise samples flagged as potentially problematic. As summarized in Table 2, the final results demonstrate the effectiveness of our approach, confirming that textual biases have been significantly suppressed, thus ensuring greater reliability and fairness of our evaluation.

Table 2: **Accuracy of the debiasing test**. Through multiple rounds of testing and revision, potential biases were effectively suppressed, ensuring the validity and reliability of MME-VideoOCR.

| Model | Visual Input | Containment Match | GPT-Assisted Scoring | Multiple-Choice |
|---|---|---|---|---|
| Qwen2.5-VL | None | 0% | 0% | 25.1% |
| Qwen2.5-VL | Black Image | 0% | 0% | 27.4% |

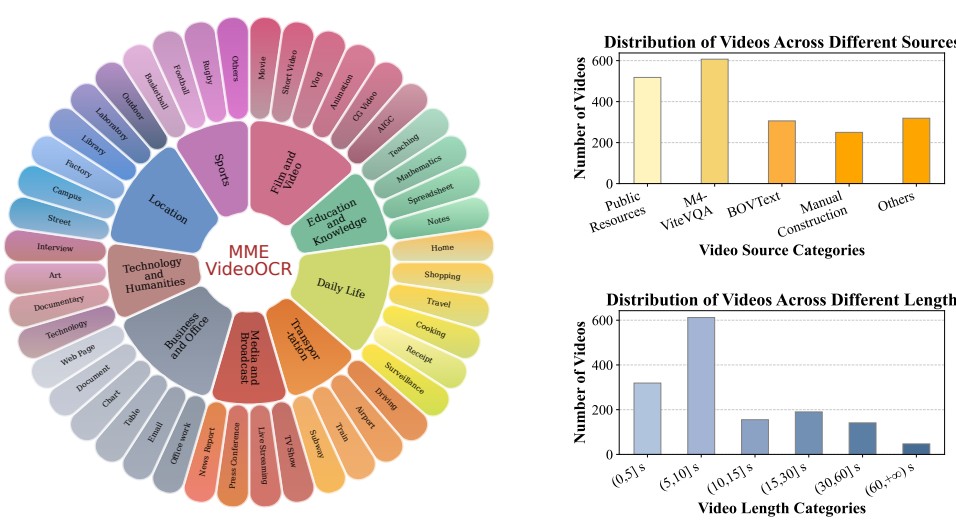

Figure 4: **Overview of MME-VideoOCR Statistics**. The videos in MME-VideoOCR covers 9 major scenario categories comprising 44 specific scene types, offering fine-grained coverage of diverse video contexts. The benchmark features a balanced distribution of video durations and sources, with a significant portion of the videos newly collected from public resources or manually curated.

## 3.5 Statistics

Through rigorous video selection, manual annotation, and expert-level validation, we collect a total of $1,464$ videos along with $2,000$ high-quality QA annotations. As illustrated in Figure 4, these videos span 9 major scenario categories, such as daily life, education and knowledge, and sports, encompassing 44 specific scenarios. The videos vary considerably in duration, resolution, and aspect ratio. They originate from diverse sources, with a substantial proportion newly collected from public resources or manually constructed.

## 4 Experiments

We evaluate a total of 18 mainstream MLLMs, including 3 cutting-edge closed-source models and 15 open-source models. The closed-source models involve GPT-4o [56], Gemini-2.5 Pro [57] and Gemini-1.5 Pro [5]. In selecting open-source models, we consider two factors. First, models are categorized by parameter size into small (7B/8B), medium (16B/32B/38B), and large (72B/78B) groups. Second, models are differentiated by their video processing strategies, including **(a)** sparse frame sampling (InternVL3 [31], LLaVA-OneVision [58], VITA-1.5 [2], LLaVA-Video [29], Kimi-VL [59], Qwen2.5-VL [30], Oryx-1.5 [60]), **(b)** dense sampling with token compression (VideoLLaMA 3 [61], VideoChat-Flash [47]), and **(c)** the slow-fast frame sampling approach (Slow-fast MLLM [62]). Please refer to Appendix C for details of the experimental setup.

### 4.1 Main Results

We evaluate the performance of all baseline models on MME-VideoOCR and display the accuracy for each task category and the overall accuracy, as shown in Table 3. Our observations indicate that among the 18 evaluated models, Gemini-2.5 Pro is the top performer, yet achieves an accuracy of only $73.7\%$. Concurrently, five models that demonstrate strong performance on other video understanding tasks achieved an accuracy below $50\%$ on MME-VideoOCR. This performance landscape underscores the challenging nature and discriminative capability of the MME-VideoOCR benchmark.

Table 3: **Evaluation results on MME-VideoOCR**. "TR" denotes Text Recognition, "VTQA" Visual Text QA, "TG" Text Grounding, "AR" Attribute Recognition, "CDT" Change Detection & Tracking, "STP" Special Text Parsing, "CFTU" Cross-Frame Text Understanding, "TBR" Text-Based Reasoning, "TBVU" Text-Based Video Understanding, and "RVT" Robust Video Testing. The highest accuracy of each task is in red , and the second highest is underlined.

| Model | Size | TR | VTQA | TG | AR | CDT | STP | CFTU | TBR | TBVU | RVT | Total |
|---|---|---|---|---|---|---|---|---|---|---|---|---|
| Closed-source MLLMs | | | | | | | | | | | | |
| Gemini-1.5 Pro | - | 76.7% | 77.6% | 61.5% | 64.7% | 55.0% | 74.0% | 31.3% | 68.7% | 53.5% | 68.0% | 64.9% |
| GPT-4o | - | 83.3% | 81.6% | 60.5% | 74.7% | 51.5% | 68.0% | 30.7% | 60.7% | 59.0% | 75.3% | 66.4% |
| Gemini-2.5 Pro | - | 83.0% | 91.6% | 64.5% | 74.0% | 70.0% | 84.4% | 48.7% | 74.0% | 56.5% | 72.0% | 73.7% |
| Small-scale MLLMs | | | | | | | | | | | | |
| LLaVA-OneVision | 7B | 42.0% | 50.0% | 49.0% | 54.0% | 41.0% | 46.4% | 20.0% | 45.3% | 52.0% | 60.0% | 46.0% |
| VideoChat-Flash | 7B | 36.7% | 48.0% | 60.0% | 60.0% | 49.0% | 46.0% | 19.3% | 50.0% | 54.0% | 60.7% | 47.8% |
| Slow-fast MLLM | 7B | 46.0% | 54.8% | 52.0% | 60.0% | 47.0% | 48.0% | 20.0% | 43.3% | 48.5% | 54.0% | 47.8% |
| VITA-1.5 | 7B | 49.0% | 58.4% | 43.0% | 61.3% | 49.0% | 53.2% | 20.0% | 51.3% | 47.0% | 58.7% | 49.5% |
| Oryx-1.5 | 7B | 51.7% | 54.0% | 50.5% | 54.7% | 44.5% | 52.8% | 23.3% | 48.7% | 47.0% | 64.0% | 49.6% |
| LLaVA-Video | 7B | 47.0% | 59.2% | 61.0% | 68.7% | 48.5% | 50.0% | 21.3% | 47.3% | 56.5% | 68.7% | 52.8% |
| VideoLLaMA 3 | 7B | 47.3% | 57.6% | 68.0% | 64.7% | 50.0% | 54.0% | 21.3% | 48.7% | 55.0% | 67.3% | 53.5% |
| Qwen2.5-VL | 7B | 70.3% | 70.0% | 58.0% | 68.7% | 48.5% | 66.4% | 17.3% | 49.3% | 53.0% | 71.3% | 59.1% |
| InternVL3 | 8B | 61.3% | 72.0% | 60.0% | 69.3% | 56.5% | 62.4% | 23.3% | 57.3% | 55.0% | 71.3% | 59.8% |
| Middle-scale MLLMs | | | | | | | | | | | | |
| Oryx-1.5 | 32B | 50.3% | 60.0% | 63.5% | 62.7% | 46.0% | 60.4% | 21.3% | 54.7% | 61.0% | 68.0% | 55.2% |
| Kimi-VL | 16B | 54.7% | 66.4% | 59.0% | 62.7% | 48.0% | 57.6% | 23.3% | 56.7% | 57.5% | 71.3% | 56.2% |
| Qwen2.5-VL | 32B | 58.3% | 77.2% | 62.5% | 68.7% | 52.0% | 70.4% | 22.7% | 68.7% | 54.5% | 65.3% | 61.0% |
| InternVL3 | 38B | 67.0% | 76.8% | 65.0% | 76.0% | 61.0% | 69.6% | 24.7% | 76.0% | 61.5% | 76.7% | 66.1% |
| Large-scale MLLMs | | | | | | | | | | | | |
| InternVL3 | 78B | 70.0% | 77.6% | 67.5% | 76.0% | 65.5% | 71.6% | 24.7% | 77.3% | 57.0% | 75.3% | 67.2% |
| Qwen2.5-VL | 72B | 80.7% | 80.0% | 65.0% | 74.0% | 56.5% | 79.6% | 26.7% | 74.7% | 57.0% | 78.7% | 69.0% |

Next, it is clear that models with larger parameter scales tend to achieve higher accuracy, with a clear scaling effect evident in the Qwen2.5-VL [30], InternVL3 [31], and Oryx-1.5 [60] series. Meanwhile, model architecture significantly impacts performance. Despite achieving high scores on general video understanding multiple-choice benchmarks (e.g., Video-MME [51], MLVU [63]), token compression methods show a clear disadvantage on MME-VideoOCR. Representative approaches such as VideoChat-Flash [47] and Slow-fast MLLM [62] illustrate this limitation, suggesting that critical information may be lost during the token merging process.

In addition, our benchmark presents strong discriminative power across task categories, which could be taken as the potential direction for MLLM optimization. For tasks such as Text Recognition, Visual Text QA, and Text-Based Reasoning, the performance gap between the best and worst-performing models exceeds 30%, clearly distinguishing model capabilities across different levels of perception, understanding, and reasoning.

Furthermore, the benchmark reveals several common defects of the mainstream MLLMs. In tasks such as Change Detection & Tracking and Text-Based Video Understanding, most models achieve an accuracy below 60%, indicating significant challenges in dynamic scene comprehension and temporal alignment. For Cross-Frame Text Understanding, which requires multi-frame integration and memory, the baseline models generally achieve an accuracy below 25%, underscoring their limited capacity for semantic integration across frames.

## 4.2 Analysis and Findings

Table 4 presents the accuracy of the top-5 performing models among the 18 evaluated MLLMs on each task. The full results for all models are provided in Appendix C.3.

**Resolution and Number of Frames**. To investigate the impact of resolution and frame count on models' performance in OCR tasks, we conduct two sets of comparative experiments. For the resolution study, we use Qwen2.5-VL [30], VideoLLaMA 3 [61], and Oryx-1.5 [60], all of which support dynamic resolution settings. In this experiment, the maximum number of input frames per sample is fixed at 32. Subsequently, the original video resolution is adjusted by scaling the longer edge of each frame to 224, 336, 448, or 560 pixels. As shown in Figure 5a, increasing the input resolution consistently leads to performance improvements across all models. To analyze the effect of input frame count, we select Qwen2.5-VL [30], InternVL3 [31], LLaVA-Video [29], and Oryx-1.5 [60], as these models are equipped with relatively long context windows. As illustrated in Figure 5b, increasing the number of input frames generally leads to a notable improvement in

Table 4: **Accuracy of top-5 performing evaluated MLLMs on each task**. Fine-grained task types offer an accurate reflection of the models' capabilities and limitations across multiple dimensions.

| Task Category | Task | Gemini 2.5-Pro | Qwen2.5-VL 72B | InternVL3 78B | GPT-4o | InternVL3 38B |
|---|---|---|---|---|---|---|
| Text Recognition | Text Recognition at Designated Locations | 86.0% | 80.5% | 72.5% | 82.0% | 70.0% |
| | Text Recognition Based on Specific Attributes | 77.0% | 81.0% | 65.0% | 86.0% | 61.0% |
| Visual Text QA | Text-Centric QA | 93.5% | 83.5% | 80.0% | 84.5% | 78.0% |
| | Translation | 84.0% | 66.0% | 68.0% | 70.0% | 72.0% |
| Text Grounding | Spatial Grounding | 88.0% | 81.0% | 77.0% | 67.0% | 83.0% |
| | Temporal Grounding | 41.0% | 49.0% | 58.0% | 54.0% | 47.0% |
| Attribute Recognition | Color Recognition | 76.0% | 90.0% | 90.0% | 88.0% | 88.0% |
| | Named Entity Recognition | 84.0% | 78.0% | 74.0% | 74.0% | 76.0% |
| | Counting | 62.0% | 54.0% | 64.0% | 62.0% | 64.0% |
| Change Detection & Tracking | Change Detection | 57.0% | 44.0% | 55.0% | 46.0% | 48.0% |
| | Tracking | 83.0% | 69.0% | 76.0% | 57.0% | 74.0% |
| Special Text Parsing | Table Parsing | 92.0% | 74.0% | 56.0% | 52.0% | 60.0% |
| | Chart Parsing | 84.0% | 72.0% | 68.0% | 68.0% | 66.0% |
| | Document Parsing | 92.0% | 94.0% | 76.0% | 90.0% | 76.0% |
| | Mathematical Formula Parsing | 68.0% | 88.0% | 90.0% | 62.0% | 80.0% |
| | Handwriting Recognition | 86.0% | 70.0% | 68.0% | 68.0% | 66.0% |
| Cross-Frame Text Understanding | Scrolling Text Understanding | 70.0% | 64.0% | 70.0% | 62.0% | 70.0% |
| | Trajectory Recognition | 0.0% | 0.0% | 0.0% | 0.0% | 0.0% |
| | Scrambled Recognition | 76.0% | 16.0% | 4.0% | 30.0% | 4.0% |
| Text-Based Reasoning | Complex Reasoning | 74.0% | 74.7% | 77.3% | 60.7% | 76.0% |
| Text-Based Video Understanding | Subtitle-Based Video Understanding | 86.0% | 96.0% | 96.0% | 93.0% | 95.0% |
| | Multi-Hop Needle in A Haystack | 27.0% | 18.0% | 18.0% | 25.0% | 28.0% |
| Robust Video Testing | AIGC Videos | 88.0% | 88.0% | 84.0% | 82.0% | 88.0% |
| | Long Videos | 44.0% | 58.0% | 68.0% | 60.0% | 62.0% |
| | Adversarial Videos | 84.0% | 90.0% | 74.0% | 84.0% | 80.0% |
| Total | - | 73.7% | 69.0% | 67.2% | 66.4% | 66.1% |

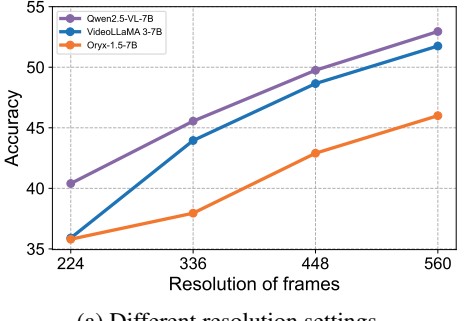

(a) Different resolution settings.

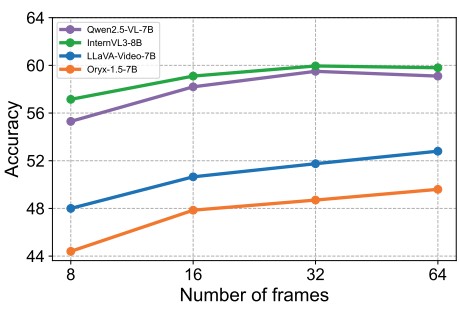

(b) Different frame sampling settings.

Figure 5: **Model performance on MME-VideoOCR under different resolution and frame sampling settings**. Both lower resolution and reduced frame count significantly degrade performance, underscoring the importance of visual coverage and clarity in OCR tasks.

model performance. However, we observe a slight performance drop for Qwen2.5-VL and InternVL3 when the number of input frames increases from 32 to 64. This suggests that when the context becomes excessively long, the models may struggle to focus on task-relevant content, potentially due to limitations in attention allocation or memory compression within long sequences. These findings highlight the importance of both high resolution and sufficient temporal coverage for OCR tasks.

**Effective Utilization of Textual Information**. In *Subtitle-Based Video Understanding*, most models achieve relatively strong performance. We investigate the task samples and reveal that the correct answers typically appear in a single frame or a small number of frames within the video. This suggests that leading MLLMs are capable of effectively utilizing textual information embedded in videos, and can combine it with visual context to perform accurate video understanding.

**Limitations in Temporal Integration Capability**. As shown in Table 3, all models exhibit clear shortcomings in Cross-Frame Text Understanding tasks, with most models achieving accuracies around 20%. Table 4 further breaks down the performance on individual tasks within this category. All of the top-5 performing models yield an accuracy of 0% on *Trajectory Recognition*, and 4 out of 5 achieve less than 35% accuracy on *Scrambled Recognition*. These results underscore a common deficiency in the temporal integration capability of current MLLMs. The large performance gap between the two subtasks under the Text-Based Video Understanding category further supports this observation. Both *Subtitle-Based Video Understanding* and *Multi-Hop Needle in A Haystack* require effective video understanding grounded in textual information. However, the key difference lies in the distribution of relevant content: in the former, useful information appears in just a few frames, whereas in the latter, it is scattered across multiple frames and requires the model to perform effective memory and integration. This contrast reveals a critical limitation in current MLLMs: rather than effectively aggregating information across multiple frames, most models appear to rely primarily on information from a small number of frames for video OCR.

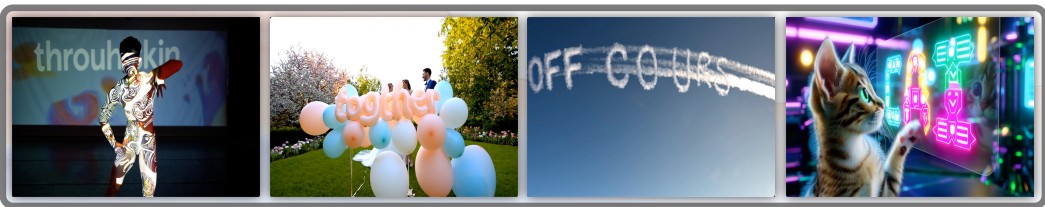

Figure 6: **Examples illustrating language prior bias in MLLMs**. The models tend to incorrectly recognize the text based on plausible language priors—for instance, "`throuh skin`" as "`through skin`", "`togther`" as "`together`", "`OFF COURS`" as "`OFF COURSE`", and "`CAI`" as "`CAT`". These cases highlight the strong influence of language priors on MLLM responses.

**Significant Language Prior Bias**. One notable failure mode in MLLMs is their tendency to over-rely on language priors when interpreting visually presented text. As illustrated in Figure 6, these models often convert visibly misspelled text into contextually plausible forms, even when the input is visually clear and unambiguous. This indicates that MLLMs frequently prioritize semantic likelihood over visual fidelity, generating interpretations that reflect linguistic expectations rather than the actual visual content. This bias poses a serious challenge for OCR-related tasks, where character-level accuracy is essential. Notably, the misrecognitions are not arbitrary; they follow consistent patterns that favor high-frequency or semantically familiar words over rare, misspelled, or out-of-vocabulary terms. Such behavior underscores the dominant role of language priors, which can override visual evidence—particularly when visual and textual signals are not sufficiently disentangled.

## 5 Conclusions, Discussions and Limitations

This paper introduces MME-VideoOCR, a benchmark designed for the comprehensive evaluation of video OCR capabilities. The benchmark comprises 25 practical OCR tasks, encompassing bilingual, perceptual, comprehension, and reasoning abilities. Experimental results demonstrate that MME-VideoOCR possesses sufficient difficulty and discrimination to expose the deficiencies of current MLLMs, thereby offering directions for the potential optimization.

While we endeavored to collect and construct videos from 9 diverse scenario categories with manually annotated, precise ground truth, the inherent richness of visual elements in videos means that some concepts may be underrepresented by samples. This may lead to score fluctuations in certain subcategories due to model sensitivity to sparse data. Although augmenting the dataset with more samples could mitigate this, we constrained the total number of items to 2,000 due to considerations of annotation and evaluation costs. Furthermore, to assess fundamental abilities, we deliberately structured the questions into easy, medium, and hard difficulty tiers. Cutting-edge MLLMs have demonstrated strong performance on the easy and medium-difficulty questions. We intend to supplement the current version with more challenging samples as MLLM capabilities advance, ensuring its continued relevance in guiding future development.

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

# A Representative Examples from MME-VideoOCR

To comprehensively illustrate the characteristics of tasks in MME-VideoOCR, we present one representative example for each task.

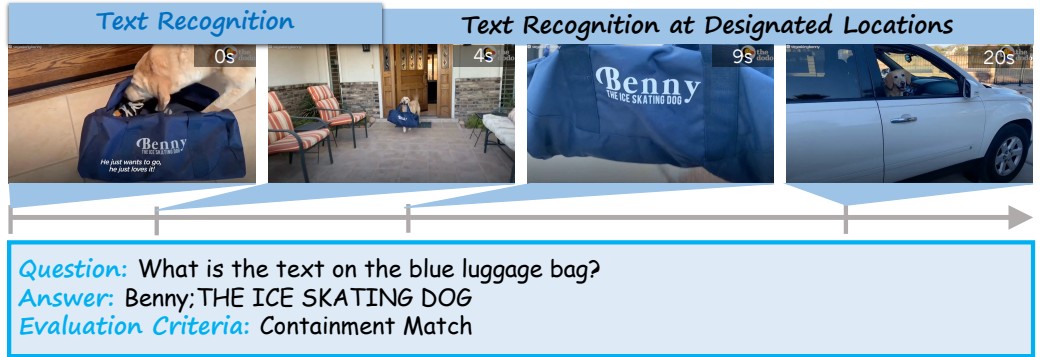

Figure 7: An example QA of the Text Recognition at Designated Locations task in MME-VideoOCR.

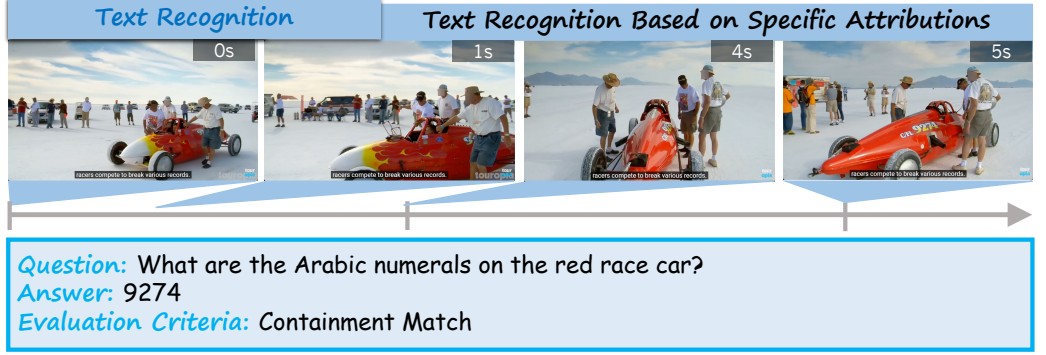

Figure 8: An example QA of the Text Recognition Based on Specific Attributes task in MME-VideoOCR.

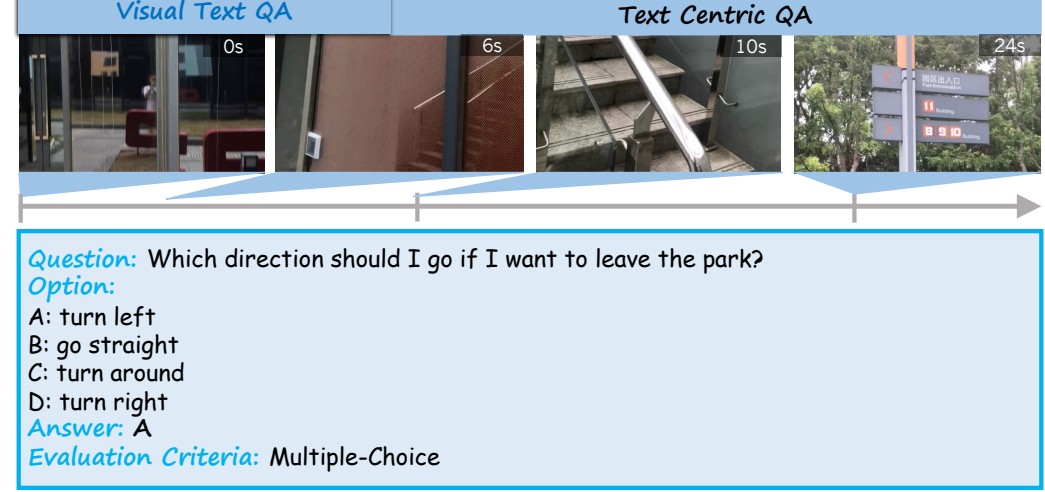

Figure 9: An example QA of the Text-Centric QA task in MME-VideoOCR.

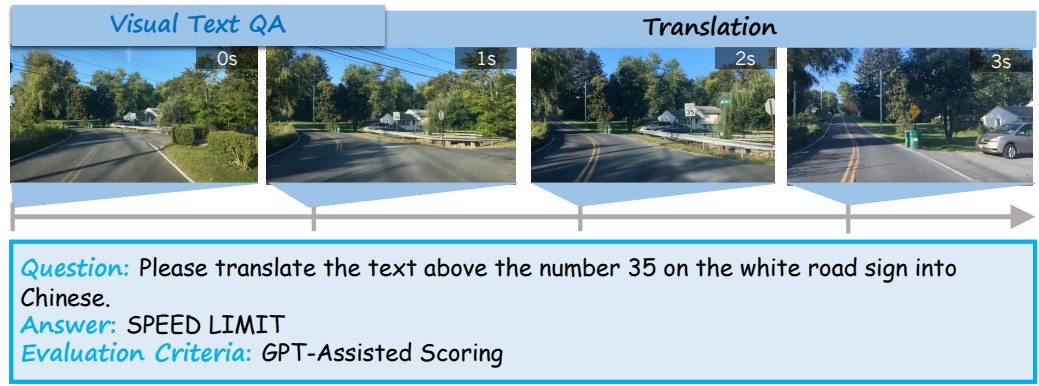

Figure 10: An example QA of the Translation task in MME-VideoOCR.

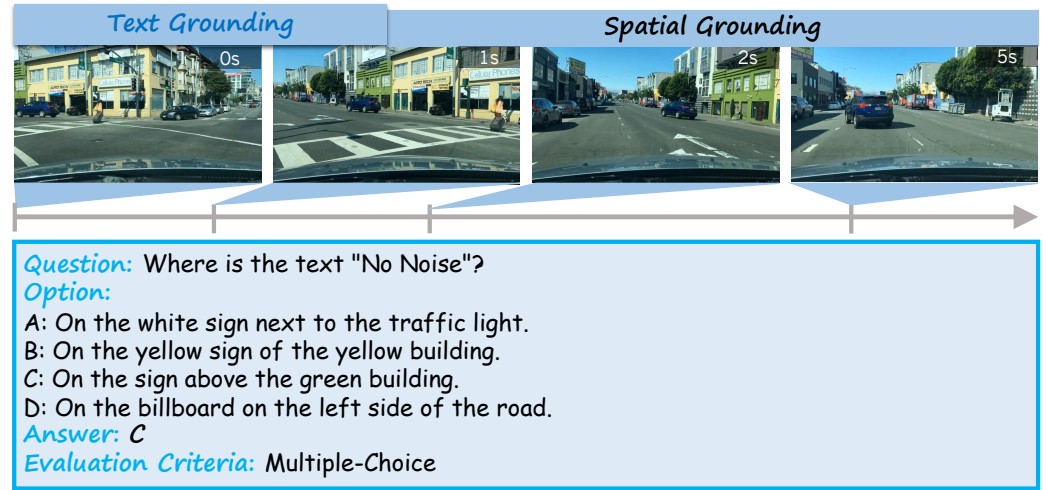

Figure 11: An example QA of the Spatial Grounding task in MME-VideoOCR.

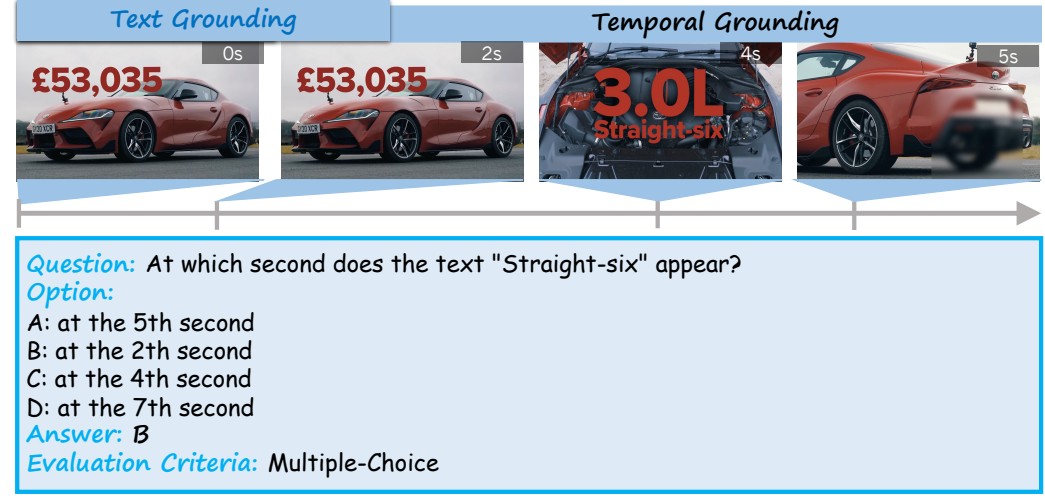

Figure 12: An example QA of the Temporal Grounding task in MME-VideoOCR.

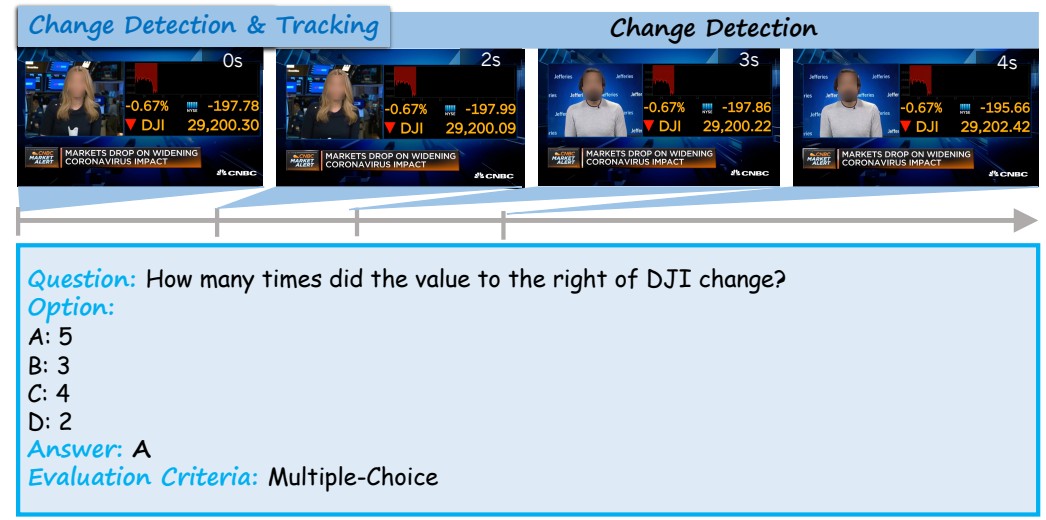

Figure 13: An example QA of the Change Detection task in MME-VideoOCR.

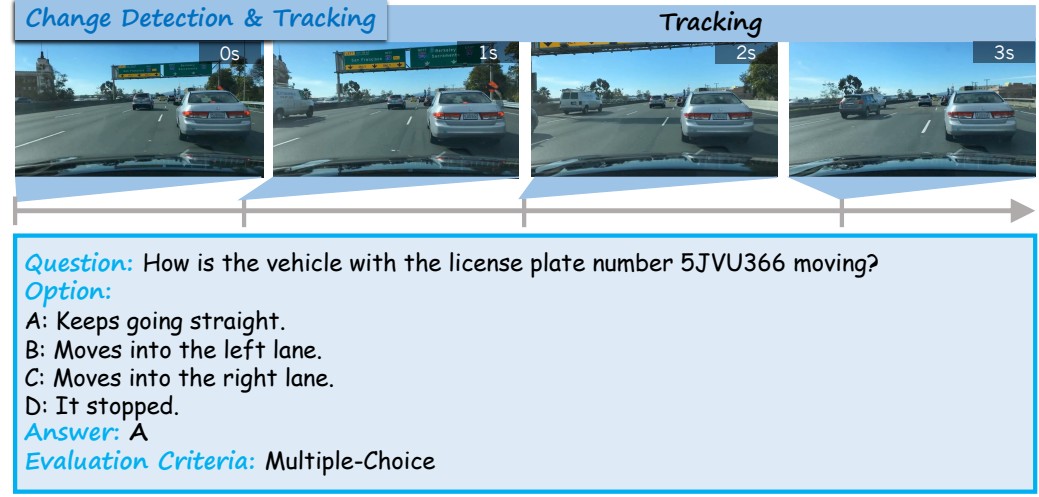

Figure 14: An example QA of the Tracking task in MME-VideoOCR.

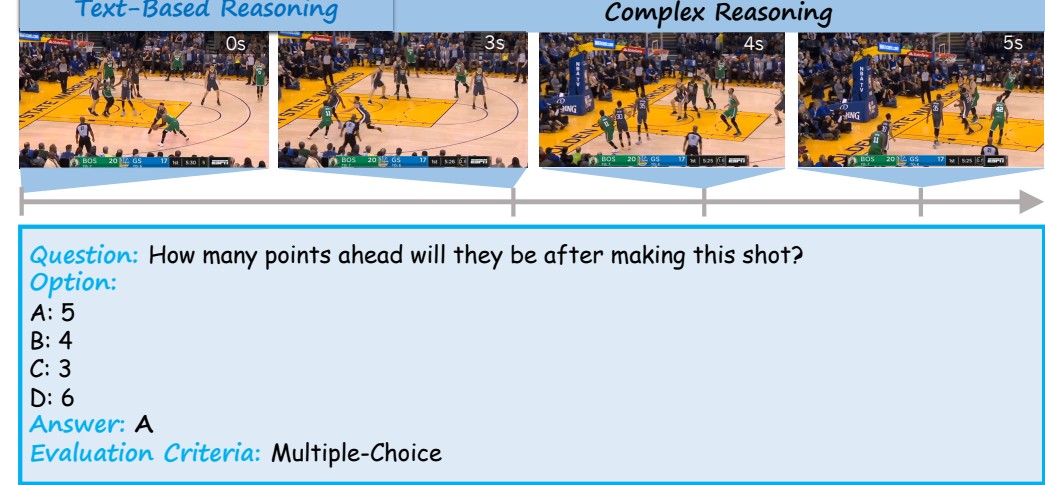

Figure 15: An example QA of the Complex Reasoning task in MME-VideoOCR.

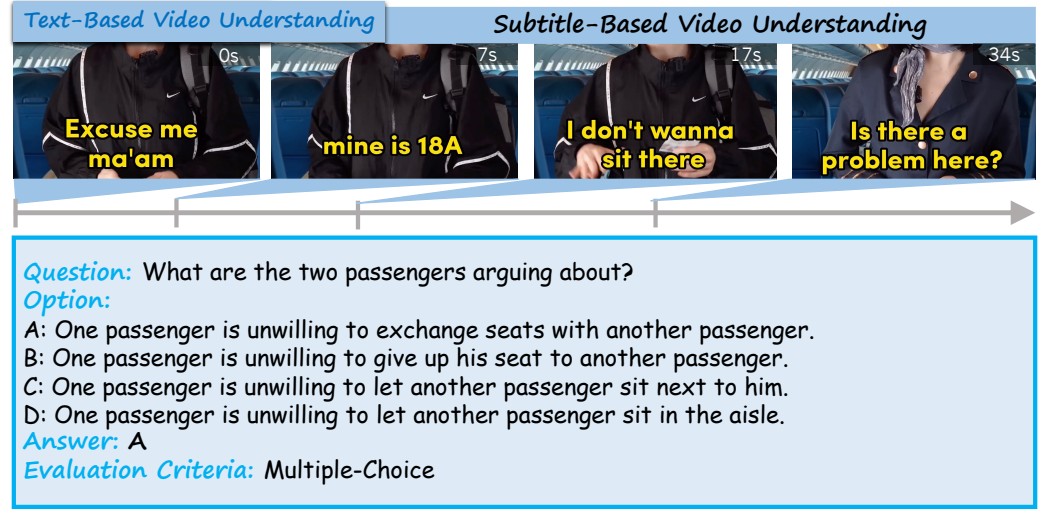

Figure 16: An example QA of the Subtitle-Based Video Understanding task in MME-VideoOCR.

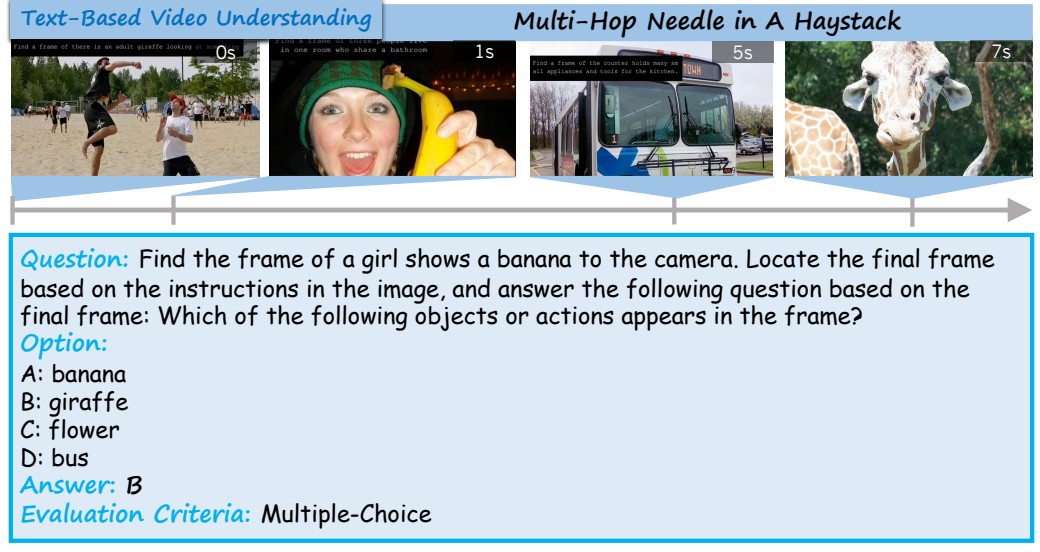

Figure 17: An example QA of the Multi-Hop Needle in A Haystack task in MME-VideoOCR.

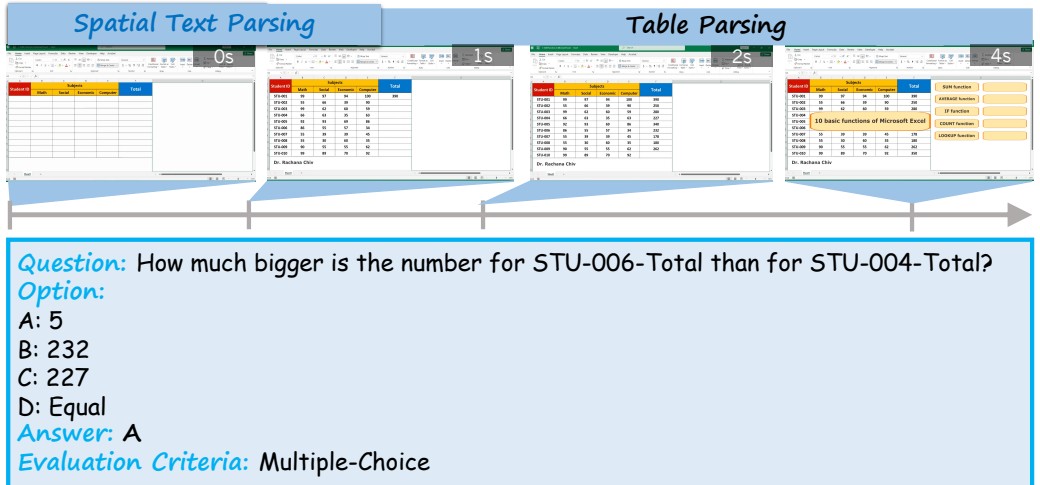

Figure 18: An example QA of the Table Parsing task in MME-VideoOCR.

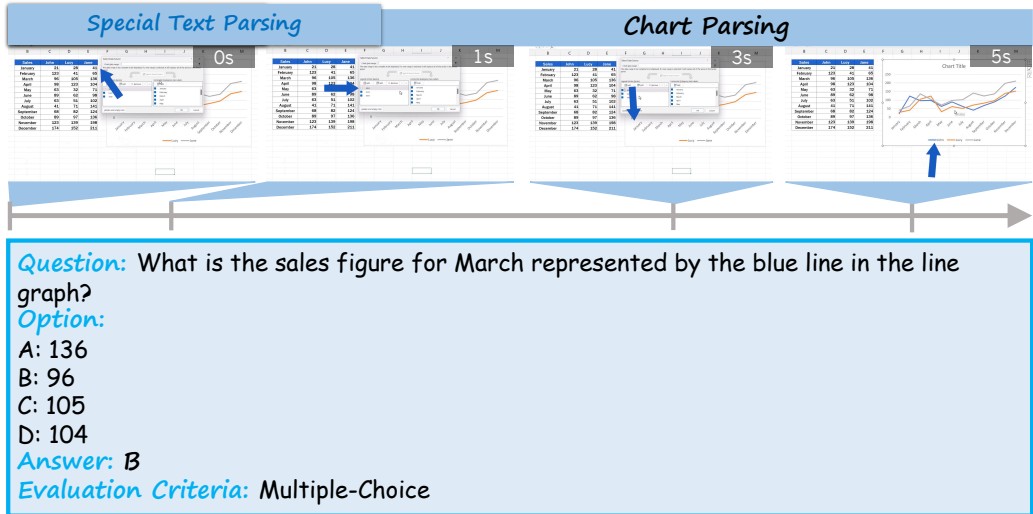

Figure 19: An example QA of the Chart Parsing task in MME-VideoOCR.

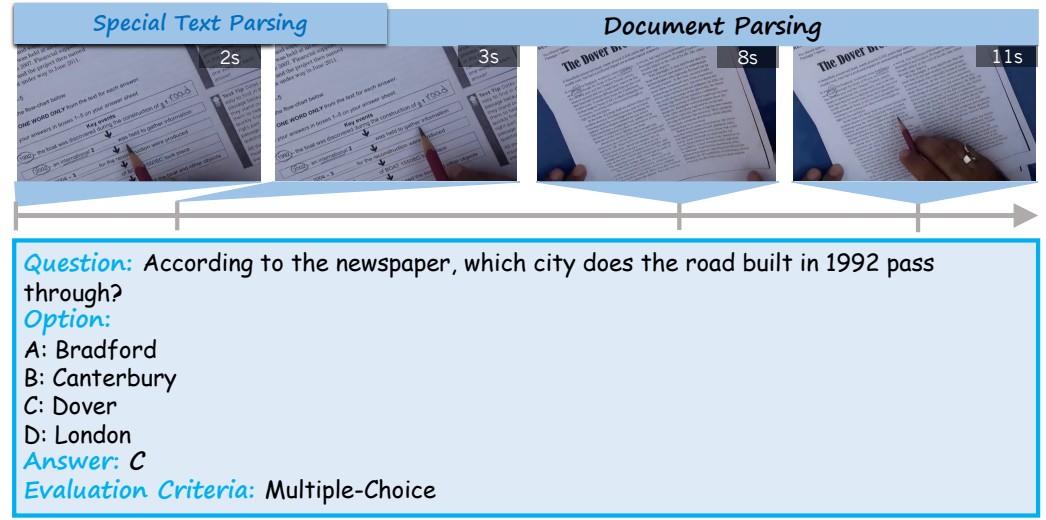

Figure 20: An example QA of the Document Parsing task in MME-VideoOCR.

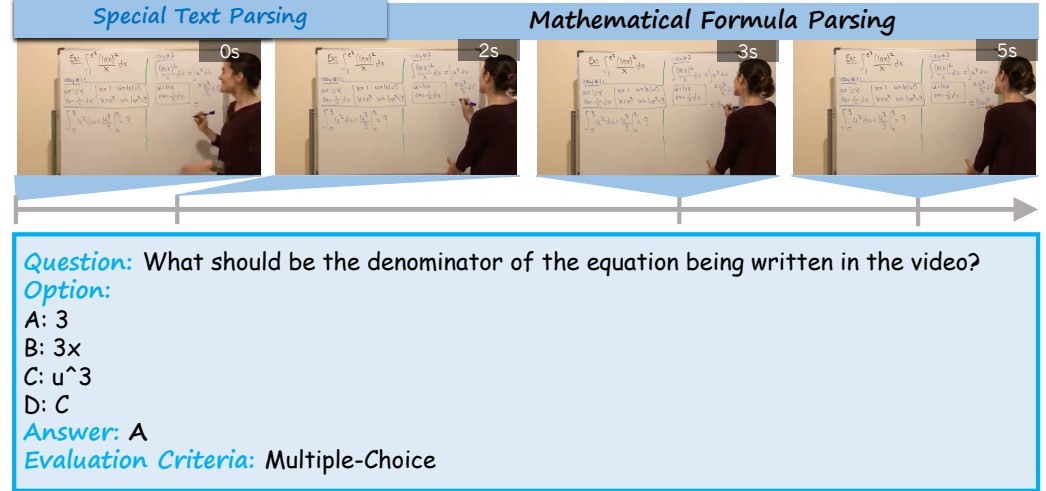

Figure 21: An example QA of the Mathematical Formula Parsing task in MME-VideoOCR.

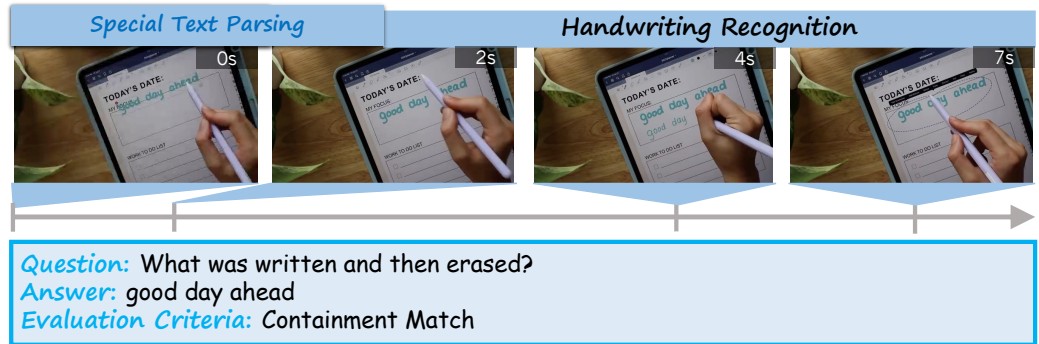

Figure 22: An example QA of the Handwriting Recognition task in MME-VideoOCR.

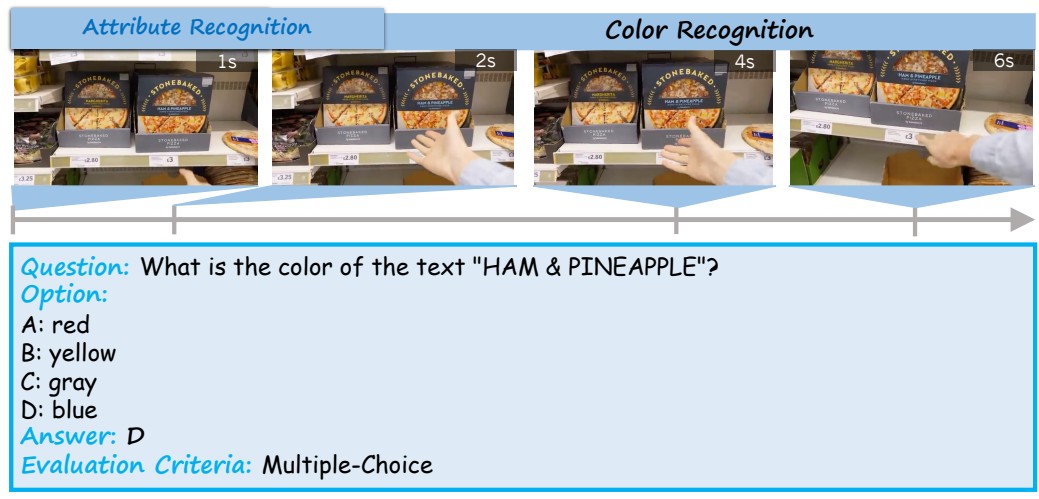

Figure 23: An example QA of the Color Recognition task in MME-VideoOCR.

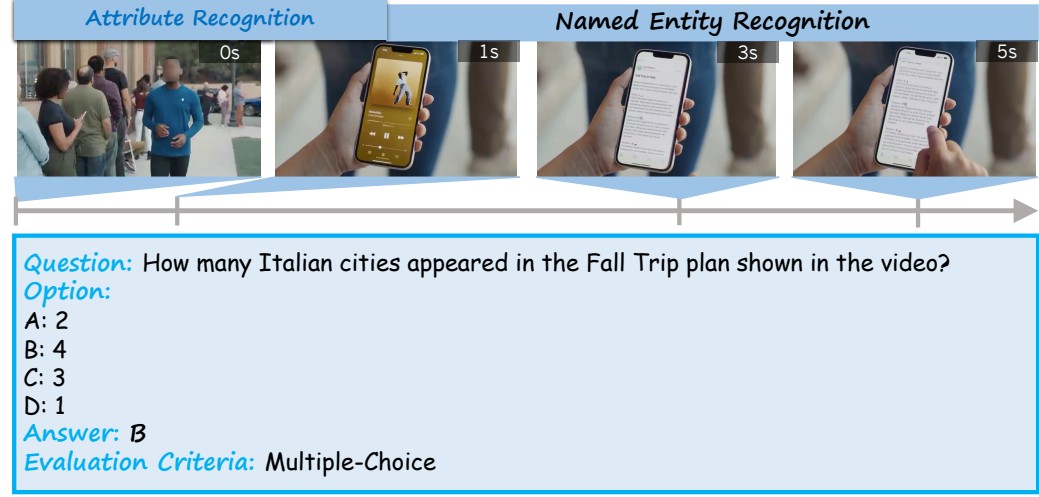

Figure 24: An example QA of the Named Entity Recognition task in MME-VideoOCR.

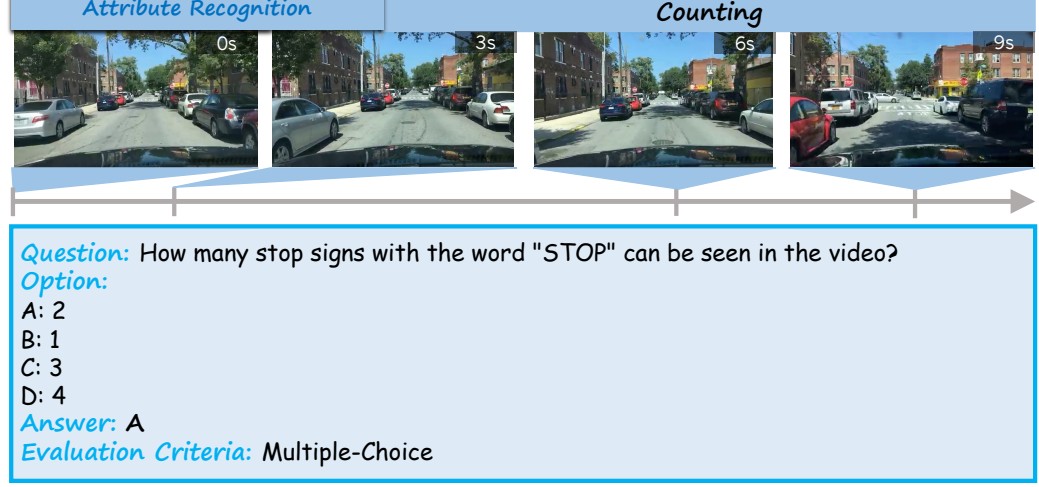

Figure 25: An example QA of the Counting task in MME-VideoOCR.

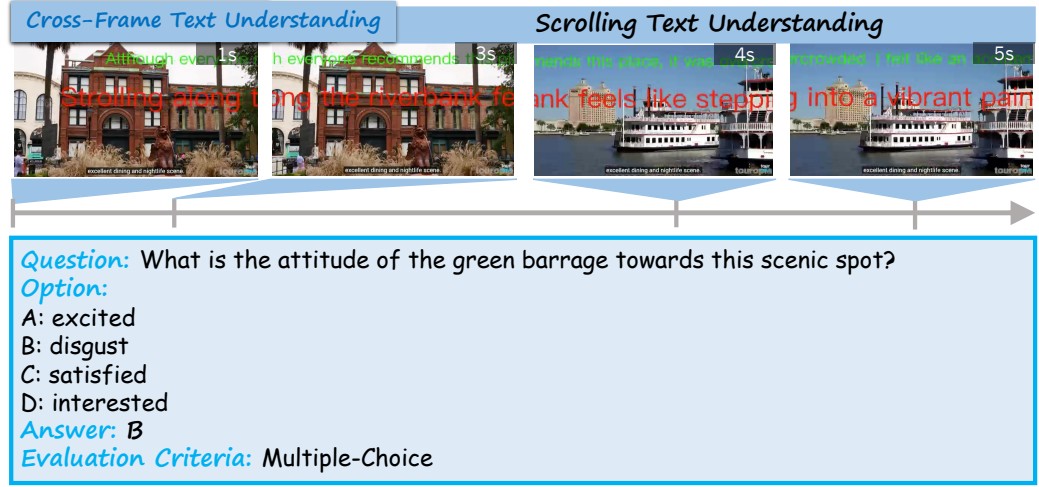

Figure 26: An example QA of the Scrolling Text Understanding task in MME-VideoOCR.

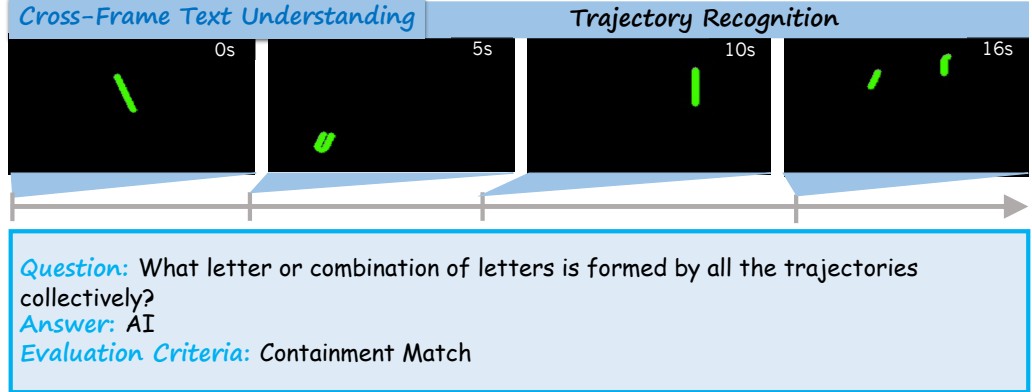

Figure 27: An example QA of the Trajectory Recognition task in MME-VideoOCR.

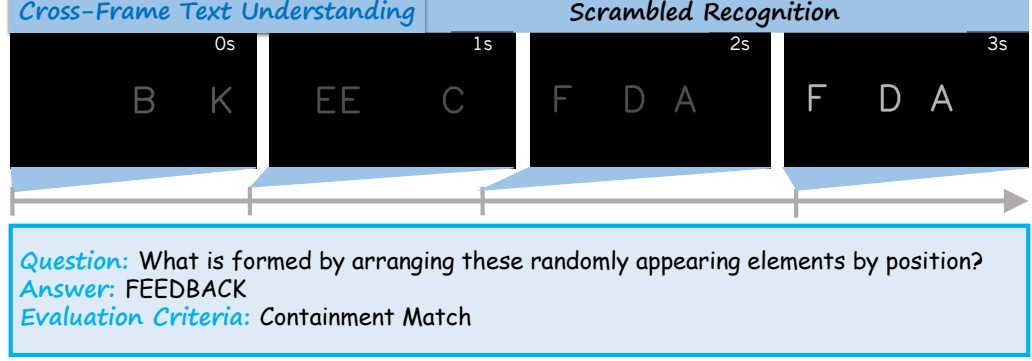

Figure 28: An example QA of the Scrambled Recognition task in MME-VideoOCR.

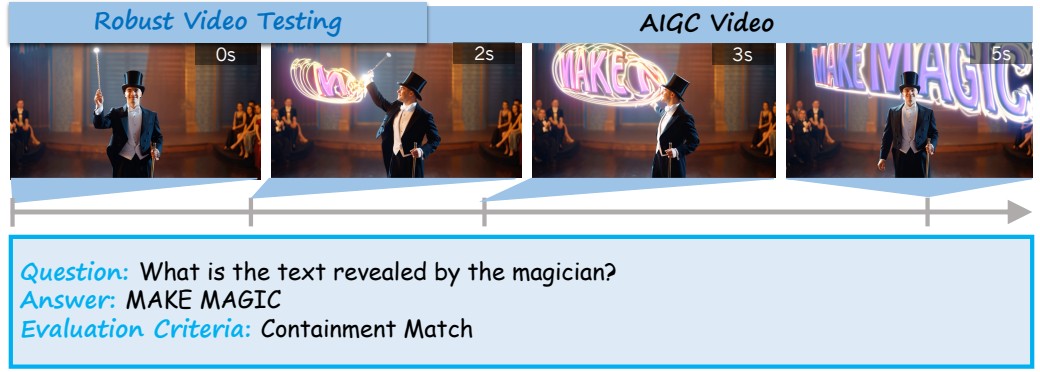

Figure 29: An example QA of the AIGC Video task in MME-VideoOCR.

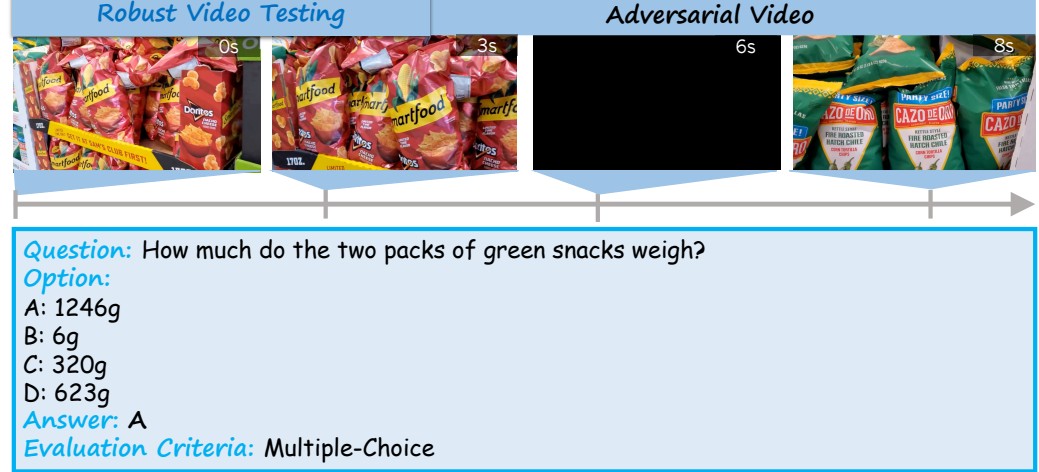

Figure 30: An example QA of the Adversarial Video task in MME-VideoOCR.

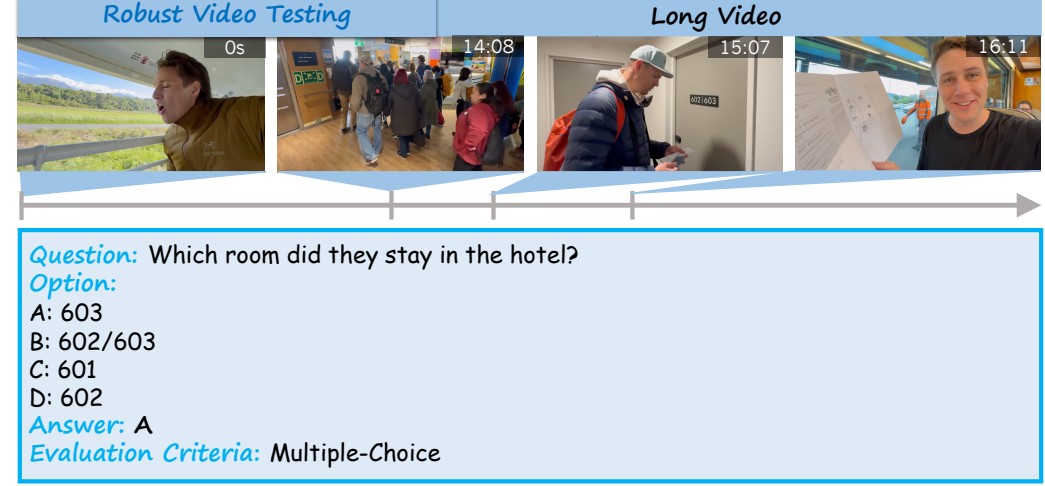

Figure 31: An example QA of the Long Video task in MME-VideoOCR.

# B  Benchmark Details

## B.1  Task Definition

MME-VideoOCR collects 10 OCR task categories. Detailed definition of the taxonomy is depicted as below.

**Text Recognition**. Text Recognition is a fundamental OCR task that evaluates an MLLM's ability to perceive and interpret text. This category involves *Text Recognition at Designated Locations* and *Text Recognition Based on Specific Attributes*. These two subtasks can be flexibly combined to assess an MLLM's capacity for fine-grained text recognition. For instance, a query may require recognizing text specifically located on a license plate and written in a particular language or color, thereby evaluating both spatial awareness and attribute-based recognition within complex visual scenes.

**Visual Text QA**. Visual Text QA encompasses two tasks: *Text-Centric QA* and *Translation*. *Text-Centric QA* requires models to integrate textual content with relevant visual cues to answer context-dependent questions. *Translation* focuses on converting specific on-screen text into a designated target language. Both tasks challenge the model's ability to not only perceive but also comprehend multimodal information.

**Text Grounding**. Text Grounding involves *Spatial Grounding* and *Temporal Grounding*. *Spatial Grounding* concerns identifying the location of specified text based on visual context—such as recognizing that the text appears on a street sign or a product label—rather than relying on exact coordinates. *Temporal Grounding* centers on understanding the temporal properties of text, including when it appears, how long it remains visible, and the sequence in which it occurs. Together, these subtasks assess the model's ability to localize and interpret text across both spatial and temporal dimensions within dynamic visual scenes.

**Attribute Recognition**. This category is composed of three tasks: *Color Recognition*, where models are expected to identify the color of the text; *Named Entity Recognition*, which focuses on extracting and classifying named entities such as person names, organization names, and location names; and *Counting*, where models must accurately identify the number of textual elements that meet specified criteria.

**Change Detection & Tracking**. The task consists of two tasks: *Change Detection* and *Tracking*. Given the highly dynamic nature of text in video, *Change Detection* aims to accurately identify changes in textual content over time. *Tracking*, on the other hand, focuses on monitoring text elements as they change position across frames—for example, tracing the movement of a vehicle with a specified license plate number or identifying the player running with the ball based on their jersey number.

**Special Text Parsing**. Special Text Parsing includes five tasks: *Table Parsing*, *Chart Parsing*, *Document Parsing*, *Mathematical Formula Parsing*, and *Handwriting Recognition*. These tasks require models to accurately identify and understand text with either special structures or highly variable visual forms.

**Cross-Frame Text Understanding**. In video scenarios, relying on a single frame is often insufficient, as critical information may be distributed across multiple frames and closely interrelated. To address this, the task of Cross-Frame Text Understanding is introduced, which requires models to integrate information across multiple frames for coherent interpretation. It includes three subtasks: *Scrolling Text Understanding*, which focuses on recognizing dynamic text streams—such as on-screen bullet comments—that move across frames and may only be fully readable when aggregated over time; *Trajectory Recognition*, where the motion path of an object in the video forms a recognizable text, and the model must interpret this trajectory as the intended message; *Scrambled Recognition*, which involves identifying and reconstructing a complete text from characters that appear out of order across different positions in the video frames.

**Text-Based Reasoning**. Text-Based Reasoning, also referred to as *Complex Reasoning*, emphasizes advanced understanding of textual content, such as code analysis, mathematical operations, and logical reasoning. Unlike *Text-Centric QA*, which is a straightforward comprehension task centered on identifying explicit information, *Complex Reasoning* requires models to go beyond surface-level understanding by synthesizing dispersed textual cues, identifying implicit relationships, and resolving ambiguity or misleading content.

**Text-Based Video Understanding**. Current video understanding tasks are primarily based on visual dynamics, such as action recognition and video captioning. However, these tasks often overlook the textual information in videos, even though they are essential for video understanding in certain contexts. To address this gap, we introduce *Subtitle-Based Video Understanding*. In this task, the answer to a question is hidden in the subtitles, and MLLMs must combine subtitle information with visual content to answer correctly. This reflects real-world scenarios like conversations, tutorials, or news, where subtitles provide key information that visuals alone cannot capture. *Multi-Hop Needle in A Haystack* is a novel and effective task introduced in VideoChat-Flash [47], designed to test models' ability to retrieve information from videos based on subtitles that are spread across multiple frames. It requires reasoning over multiple pieces of subtitle content to find the correct answer.

**Robust Video Testing**. To evaluate model effectiveness and robustness across diverse scenarios, we introduce three specialized video types: *AIGC Videos*, *Long Videos*, and *Adversarial Videos*. *AIGC Videos*, generated by AI systems [42], assess model adaptability to increasingly common synthetic content. *Long Videos* test the ability to extract relevant information from lengthy sequences with substantial redundancy. Since existing MLLMs primarily process videos by extracting frames, we construct a set of *Adversarial Videos* by strategically inserting all-black frames into normal videos. While these adversarial samples have minimal impact on human perception, they can easily mislead the model, rendering it virtually "blind".

## B.2 Task Distribution

Table 5: **Number of QA Pairs per task in MME-VideoOCR**.

| Task Category | Task | #QA |
|---|---|---|
| Text Recognition | Text Recognition at Designated Locations | 200 |
| | Text Recognition Based on Specific Attributes | 100 |
| Visual Text QA | Text-Centric QA | 250 |
| | Translation | 50 |
| Text Grounding | Spatial Grounding | 100 |
| | Temporal Grounding | 100 |
| Attribute Recognition | Color Recognition | 50 |
| | Named Entity Recognition | 50 |
| | Counting | 50 |
| Change Detection & Tracking | Change Detection | 100 |
| | Tracking | 100 |
| Special Text Parsing | Table Parsing | 50 |
| | Chart Parsing | 50 |
| | Document Parsing | 50 |
| | Mathematical Formula Parsing | 50 |
| | Handwriting Recognition | 50 |
| Cross-Frame Text Understanding | Scrolling Text Understanding | 50 |
| | Trajectory Recognition | 50 |
| | Scrambled Recognition | 50 |
| Text-Based Reasoning | Complex Reasoning | 150 |
| Text-Based Video Understanding | Subtitle-Based Video Understanding | 100 |
| | Multi-Hop Needle in a Haystack | 100 |
| Robust Video Testing | AIGC Videos | 50 |
| | Long Videos | 50 |
| | Adversarial Videos | 50 |
| Total | - | 2,000 |

Given the diverse range of task types included in MME-VideoOCR, which assess a broad spectrum of model capabilities, we carefully allocate the number of QA pairs across different tasks. Table 5

Table 6: **Evaluation prompt setting of MME-VideoOCR (Containment Match)**.

[Video]
Based on the video and the question below, directly answer the content that needs to be recognized in plain text. Do not include any additional explanations, formatting changes, or extra information.
Question: [Question]
The answer is:

Table 7: **Evaluation prompt setting of MME-VideoOCR (GPT-Assisted Scoring)**.

[Video]
Based on the video and the question below, directly provide the answer in plain text. Do not include any additional explanations, formatting changes, or extra information.
Question: [Question]
The answer is:

presents the specific number of QA pairs for each task. This allocation ensures a balanced distribution among perception, understanding, and reasoning tasks, thereby supporting a comprehensive and equitable evaluation of model capabilities.

### B.3 Evaluation Prompt

The prompt settings for Containment Match, GPT-Assisted Scoring and Multiple-Choice are shown in Table 6, Table 7 and Table 8. For GPT-Assisted Scoring (designed for the Translation task), after obtaining the model's response using the prompt shown in Table 7, we subsequently utilize GPT-4o-0806 to evaluate the response. The corresponding evaluation prompt is provided in Table 9.

## C   Experiment Details

### C.1   Evaluated Models

We evaluate a total of 18 mainstream MLLMs, including 3 leading proprietary models and 15 high-performing open-source models.

For proprietary models, we evaluate GPT-4o [56], Gemini-2.5 Pro [57] and Gemini-1.5 Pro [5].

- *GPT-4o* is the latest multimodal large language model developed by OpenAI, offering fast and cost-effective performance across text, image, and audio modalities. It achieves state-of-the-art results on a variety of benchmarks, with notable improvements in visual reasoning, OCR, and multilingual understanding. GPT-4o features a unified architecture that enables seamless cross-modal interaction, making it highly efficient and versatile for real-world multimodal applications.

- *Gemini-2.5 Pro* is one of the latest Multimodal Large Language Models released by Google DeepMind. It features improved visual and video understanding capabilities, with support for extended context lengths and more efficient cross-modal alignment. Gemini-2.5 Pro demonstrates strong performance across a wide range of tasks, including video captioning, image reasoning, and OCR-based understanding. Its enhanced architecture and training scale make it particularly competitive in complex multimodal benchmarks.

- *Gemini-1.5 Pro*, an earlier version in the Gemini series, also supports multimodal input and is optimized for high-quality text generation and basic vision-language tasks. While it delivers reliable performance on standard image-based benchmarks, its video comprehension ability—especially in tasks requiring temporal reasoning and dense visual-textual alignment—is more limited compared to its successor. Nevertheless, it remains a strong baseline among proprietary models.

For open-source models, we select Qwen2.5-VL [30], LLaVA-Video [29], LLaVA-OneVision [58], VideoLLaMA 3 [61], VideoChat-Flash [47], Oryx-1.5 [60], Slowfast-MLLM [62], InternVL3 [64], VITA-1.5 [2] and Kimi-VL [59]. Among them, for Oryx-1.5, Qwen2.5-VL, and InternVL3, we include versions with different parameter scales in our experiments.

Table 8: **Evaluation prompt setting of MME-VideoOCR (Multiple-Choice)**.

```
[Video]
```
Select the best answer to the following multiple-choice question based on the video. Respond with only the letter (A, B, C, or D) of the correct option.
Question: `[Question]`
Option:
A. `[Option A]`
B. `[Option B]`
C. `[Option C]`
D. `[Option D]`
The best answer is:

Table 9: **Evaluation prompt setting of the Translation task**.

You are a professional bilingual translation evaluator.

Here are two sentences: one in Chinese and one in English.
Sentence 1: `[Ground Truth]`
Sentence 2: `[MLLM's Response]`

Please evaluate whether the two sentences convey the same meaning and can be considered accurate translations of each other.

If the meanings are equivalent and the translation is accurate, respond with "correct".
If there are significant differences in meaning or inaccuracies in translation, respond with "wrong".

You must only respond with one word: "correct" or "wrong". Do not provide any explanations, comments, or additional text.
Focus solely on semantic equivalence, not grammar or style. Ignore minor differences as long as the meaning is preserved.

- *Qwen2.5-VL* is a vision-language model that introduces two key innovations: native dynamic-resolution processing and Multi-scale Rotary Position Embedding (MRoPE). The dynamic-resolution capability allows the model to process images and videos of varying resolutions and frame rates efficiently, extending to the temporal dimension through dynamic FPS sampling. This enables precise temporal event localization in long videos. MRoPE enhances the model's ability to capture multi-scale positional information, improving its performance in tasks requiring fine-grained spatial and temporal understanding .

- *LLaVA-Video* extends the LLaVA framework to video understanding by unifying visual representations into the language feature space. This alignment before projection enables the model to perform visual reasoning on both images and videos simultaneously. By training on a mixed dataset of images and videos, LLaVA-Video leverages mutual enhancement between modalities, achieving superior performance across various visual-language tasks .

- *LLaVA-OneVision* is designed for versatile visual task transfer across single-image, multi-image, and video scenarios. It employs a SigLIP vision encoder and a Qwen2 language backbone, processing images with the Anyres technique to handle high-resolution inputs effectively. Videos are processed with a fixed token length per frame for memory efficiency. This architecture enables LLaVA-OneVision to excel in diverse visual-language tasks without task-specific fine-tuning.

- *VideoLLaMA 3* is a vision-centric multimodal foundation model that advances image and video understanding. It utilizes Any-resolution Vision Tokenization (AVT) to process images and videos of varying resolutions dynamically. The model's training paradigm emphasizes high-quality image-text data to enhance video understanding capabilities. VideoLLaMA 3 achieves state-of-the-art performance on multiple benchmarks by integrating vision-centric training and framework designs.

- *VideoChat-Flash* is a long-context video-language model that introduces a Hierarchical visual token Compression (HiCo) method, effectively reducing redundancy in long videos by compressing visual tokens from the clip-level to the video-level. This approach enables high-fidelity representation while significantly lowering computational costs. Coupled with

a multi-stage short-to-long learning scheme and training on the LongVid dataset, VideoChat-Flash achieves state-of-the-art performance on both long and short video benchmarks.

- *Oryx-1.5* presents a unified multimodal architecture designed for on-demand spatial-temporal understanding of images, videos, and multi-view 3D scenes. It features a dynamic compressor module that performs token compression and adaptive positional embedding, allowing the model to efficiently process visual inputs with arbitrary spatial sizes and temporal lengths. This flexibility enables Oryx-1.5 to seamlessly handle diverse visual inputs across various modalities.

- *Slowfast-MLLM* integrates the SlowFast dual-pathway architecture with a multimodal large language model to explicitly capture both coarse and fine-grained temporal dynamics. The slow branch models long-term context, while the fast branch focuses on short-term changes, enabling rich motion representation. This design enhances temporal alignment and supports detailed video-text interaction in tasks such as action question answering and event tracking.

- *InternVL3* is a powerful vision-language model that unifies visual grounding, dense captioning, and temporal understanding via a cross-modality fusion backbone. It introduces region-level supervision and multi-frame alignment strategies, significantly improving its spatial-temporal grounding capabilities. InternVL3 demonstrates superior performance across a wide range of multimodal tasks, benefiting from its native multimodal pre-training paradigm and advanced post-training techniques.

- *VITA-1.5* is a multimodal large language model designed to achieve real-time vision and speech interaction. It pioneers a meticulously crafted three-stage training strategy to effectively integrate vision, language, and speech modalities. This strategy systematically introduces visual and auditory data, mitigating conflicts between modalities while preserving robust multimodal capabilities. This methodology empowers VITA-1.5 to process and understand both visual and speech inputs and to generate fluent, end-to-end speech outputs, thereby enabling more natural and seamless interactive multimodal conversations.

- *Kimi-VL* is a state-of-the-art vision-language model developed by Moonshot AI, based on the Kimi series of large language models. Designed to handle complex multimodal tasks, Kimi-VL integrates high-resolution visual encoders with large-scale language understanding to enable robust performance in image captioning, visual question answering, and document understanding. It adopts a Mixture-of-Experts (MoE) architecture to improve inference efficiency, dynamically activating a subset of experts for each input. This design allows Kimi-VL to scale effectively while maintaining strong generalization across diverse visual-language benchmarks.

## C.2 Experimental Setup

For proprietary models, we used the `gpt-4o-2024-08-06`, `gemini-2.5-pro-preview-05-06` and `gemini-1.5-pro-002` APIs, respectively.

In the MME-VideoOCR evaluation, most models were configured with a maximum input frame count of 64. GPT-4o was limited to 50 input frames due to API token constraints, while VITA-1.5 was restricted to 16 frames because of context length limitations. All other settings followed default or recommended configurations.
During the comparative experiments described in Section 4.2, the number of input frames was fixed at 32 when varying the resolution, while the default resolution settings were applied to all models when varying the number of input frames.

## C.3 Experiment Results

Table 10, Table 11 and Table 12 present the complete results of evaluated models across all tasks in MME-VideoOCR.

# D Impact Statement

This paper presents work whose goal is to advance the field of Machine Learning. There are many potential societal consequences of our work, none which we feel must be specifically highlighted here.

Table 10: **Accuracy of evaluated MLLMs on each task of MME-VideoOCR.**

| Task Category | Task | Gemini 1.5 Pro | Qwen2.5-VL 32B | InternVL 8B | Qwen2.5-VL 7B | Kimi-VL |
|---|---|---|---|---|---|---|
| Text Recognition | Text Recognition at Designated Locations | 80.0% | 55.0% | 64.0% | 70.0% | 54.5% |
| | Text Recognition Based on Specific Attributes | 70.0% | 65.0% | 56.0% | 71.0% | 55.0% |
| Visual Text QA | Text-Centric QA | 83.0% | 81.5% | 75.5% | 76.0% | 68.5% |
| | Translation | 56.0% | 60.0% | 58.0% | 46.0% | 58.0% |
| Text Grounding | Spatial Grounding | 78.0% | 73.0% | 77.0% | 77.0% | 71.0% |
| | Temporal Grounding | 45.0% | 52.0% | 43.0% | 39.0% | 47.0% |
| Attribute Recognition | Color Recognition | 62.0% | 78.0% | 80.0% | 78.0% | 70.0% |
| | Named Entity Recognition | 80.0% | 78.0% | 72.0% | 76.0% | 70.0% |
| | Counting | 52.0% | 50.0% | 56.0% | 52.0% | 48.0% |
| Change Detection & Tracking | Change Detection | 43.0% | 40.0% | 49.0% | 40.0% | 33.0% |
| | Tracking | 67.0% | 64.0% | 64.0% | 57.0% | 63.0% |
| Special Text Parsing | Table Parsing | 72.0% | 66.0% | 56.0% | 58.0% | 54.0% |
| | Chart Parsing | 74.0% | 60.0% | 60.0% | 68.0% | 48.0% |
| | Document Parsing | 80.0% | 90.0% | 72.0% | 86.0% | 74.0% |
| | Mathematical Formula Parsing | 76.0% | 76.0% | 64.0% | 60.0% | 60.0% |
| | Handwriting Recognition | 68.0% | 60.0% | 60.0% | 60.0% | 52.0% |
| Cross-Frame Text Understanding | Scrolling Text Understanding | 72.0% | 52.0% | 70.0% | 48.0% | 70.0% |
| | Trajectory Recognition | 0.0% | 0.0% | 0.0% | 0.0% | 0.0% |
| | Scrambled Recognition | 22.0% | 16.0% | 0.0% | 4.0% | 0.0% |
| Text-Based Reasoning | Complex Reasoning | 68.7% | 68.7% | 57.3% | 49.3% | 56.7% |
| Text-Based Video Understanding | Subtitle-Based Video Understanding | 90.0% | 93.0% | 96.0% | 90.0% | 95.0% |
| | Multi-Hop Needle in A Haystack | 17.0% | 16.0% | 14.0% | 16.0% | 20.0% |
| Robust Video Testing | AIGC Videos | 86.0% | 66.0% | 86.0% | 78.0% | 82.0% |
| | Long Videos | 42.0% | 46.0% | 50.0% | 56.0% | 54.0% |
| | Adversarial Videos | 76.0% | 84.0% | 78.0% | 80.0% | 78.0% |
| Total | - | 64.9% | 61.0% | 59.8% | 59.1% | 56.2% |

Table 11: **Accuracy of evaluated MLLMs on each task of MME-VideoOCR**.

| Task Category | Task | Oryx-1.5 32B | Video-LLaMA 3 | LLaVA Video-7B | Oryx-1.5 7B |
|---|---|---|---|---|---|
| Text Recognition | Text Recognition at Designated Locations | 52.5% | 47.5% | 49.0% | 53.0% |
| | Text Recognition Based on Specific Attributes | 46.0% | 47.0% | 43.0% | 49.0% |
| Visual Text QA | Text-Centric QA | 67.0% | 63.5% | 67.0% | 62.0% |
| | Translation | 32.0% | 34.0% | 28.0% | 22.0% |
| Text Grounding | Spatial Grounding | 73.0% | 65.0% | 70.0% | 59.0% |
| | Temporal Grounding | 54.0% | 71.0% | 52.0% | 42.0% |
| Attribute Recognition | Color Recognition | 66.0% | 76.0% | 84.0% | 64.0% |
| | Named Entity Recognition | 68.0% | 66.0% | 66.0% | 64.0% |
| | Counting | 54.0% | 52.0% | 56.0% | 36.0% |
| Change Detection & Tracking | Change Detection | 37.0% | 39.0% | 40.0% | 35.0% |
| | Tracking | 55.0% | 61.0% | 57.0% | 54.0% |
| Special Text Parsing | Table Parsing | 52.0% | 44.0% | 44.0% | 50.0% |
| | Chart Parsing | 46.0% | 50.0% | 42.0% | 44.0% |
| | Document Parsing | 76.0% | 68.0% | 64.0% | 70.0% |
| | Mathematical Formula Parsing | 74.0% | 64.0% | 56.0% | 58.0% |
| | Handwriting Recognition | 54.0% | 44.0% | 44.0% | 42.0% |
| Cross-Frame Text Understanding | Scrolling Text Understanding | 64.0% | 60.0% | 60.0% | 68.0% |
| | Trajectory Recognition | 0.0% | 0.0% | 0.0% | 0.0% |
| | Scrambled Recognition | 0.0% | 4.0% | 4.0% | 2.0% |
| Text-Based Reasoning | Complex Reasoning | 54.7% | 48.7% | 47.3% | 48.7% |
| Text-Based Video Understanding | Subtitle-Based Video Understanding | 86.0% | 91.0% | 93.0% | 78.0% |
| | Multi-Hop Needle in A Haystack | 36.0% | 19.0% | 20.0% | 16.0% |
| Robust Video Testing | AIGC Videos | 80.0% | 78.0% | 86.0% | 80.0% |
| | Long Videos | 52.0% | 56.0% | 54.0% | 40.0% |
| | Adversarial Videos | 72.0% | 68.0% | 66.0% | 72.0% |
| Total | - | 55.2% | 53.5% | 52.8% | 49.6% |

Table 12: **Accuracy of evaluated MLLMs on each task of MME-VideoOCR**.

| Task Category | Task | VITA-1.5 | Slow-fast MLLM | Videochat-Flash-7B | LLaVA OneVision-7B |
|---|---|---|---|---|---|
| Text Recognition | Text Recognition at Designated Locations | 48.0% | 46.0% | 37.5% | 42.0% |
| | Text Recognition Based on Specific Attributes | 51.0% | 46.0% | 35.0% | 42.0% |
| Visual Text QA | Text-Centric QA | 63.0% | 61.5% | 55.5% | 57.0% |
| | Translation | 40.0% | 28.0% | 18.0% | 22.0% |
| Text Grounding | Spatial Grounding | 53.0% | 61.0% | 61.0% | 58.0% |
| | Temporal Grounding | 33.0% | 43.0% | 59.0% | 40.0% |
| Attribute Recognition | Color Recognition | 66.0% | 66.0% | 64.0% | 66.0% |
| | Named Entity Recognition | 58.0% | 70.0% | 66.0% | 62.0% |
| | Counting | 60.0% | 44.0% | 50.0% | 34.0% |
| Change Detection & Tracking | Change Detection | 37.0% | 44.0% | 43.0% | 36.0% |
| | Tracking | 61.0% | 50.0% | 55.0% | 46.0% |
| Special Text Parsing | Table Parsing | 44.0% | 42.0% | 32.0% | 40.0% |
| | Chart Parsing | 44.0% | 42.0% | 40.0% | 40.0% |
| | Document Parsing | 72.0% | 64.0% | 56.0% | 56.0% |
| | Mathematical Formula Parsing | 64.0% | 60.0% | 58.0% | 56.0% |
| | Handwriting Recognition | 42.0% | 32.0% | 44.0% | 40.0% |
| Cross-Frame Text Understanding | Scrolling Text Understanding | 60.0% | 58.0% | 58.0% | 58.0% |
| | Trajectory Recognition | 0.0% | 0.0% | 0.0% | 0.0% |
| | Scrambled Recognition | 0.0% | 2.0% | 0.0% | 2.0% |
| Text-Based Reasoning | Complex Reasoning | 51.3% | 43.3% | 50.0% | 45.3% |
| Text-Based Video Understanding | Subtitle-Based Video Understanding | 83.0% | 83.0% | 88.0% | 86.0% |
| | Multi-Hop Needle in A Haystack | 11.0% | 14.0% | 20.0% | 18.0% |
| Robust Video Testing | AIGC Videos | 68.0% | 58.0% | 78.0% | 78.0% |
| | Long Videos | 42.0% | 38.0% | 44.0% | 36.0% |
| | Adversarial Videos | 66.0% | 66.0% | 60.0% | 66.0% |
| Total | - | 49.5% | 47.8% | 47.8% | 46.0% |

