# OpenReview forum: "MME-VideoOCR: Evaluating OCR-Based Capabilities of Multimodal LLMs in Video Scenarios"
_NeurIPS.cc/2025/Datasets_and_Benchmarks_Track — NeurIPS 2025 Datasets and Benchmarks Track poster_

### Official Review · Reviewer_kcWJ · 2025-06-07

**Rating:** 5
**Confidence:** 4

**Summary:**

This paper addresses the limitations of current Multimodal Large Language Models (MLLMs) in handling Optical Character Recognition (OCR) tasks within videos, where challenges like motion blur and temporal variation hinder performance. To tackle this, the authors introduce MME-VideoOCR, a comprehensive benchmark designed to guide the development of more robust MLLMs for video OCR.

**Dataset Code Accessibility:**

Yes

**Ethical Considerations:**

No, there are no or only very minor ethics concerns

**Final Justification:**

Thank you for your response. My concerns have been addressed so that I will maintain the original score.

**Limitations Weaknesses:**

The sample scenarios in the benchmark are relatively complex, but most of the answers provided are in a single-choice format. This limits the interpretability of model outputs. Incorporating visual grounding and a clearer exposition of the model reasoning process could enhance the explainability of results. Evaluating models on these aspects would also strengthen the overall reliability of the benchmark.

It is recommended that the authors discuss these concerns in the rebuttal. Nonetheless, this does not diminish the value of the current work, which remains a strong and commendable contribution.

**Strengths Contributions:**

1. This paper presents a highly comprehensive video OCR benchmark. Compared to previous approaches, it includes a broader range of tasks and offers a thorough evaluation of MLLMs across perception, understanding, and reasoning capabilities. Additionally, its support for bilingual content contributes significantly to the development of the field.

2. The paper is clearly written, with excellent presentation throughout. The motivation behind the benchmark's construction is clearly articulated.

3. The benchmark development process is rigorous: samples are collected from multiple data sources, annotated by humans, and verified by experts. This ensures both diversity and annotation accuracy. The bias mitigation process further enhances the benchmark's ability to reflect the true visual capabilities of MLLMs.

---

> ### Author Rebuttal · Authors · 2025-07-30
>
> Thank you for your valuable comments and suggestions! We sincerely appreciate the time and effort you devoted to reviewing our work!
>
> We agree that incorporating visual grounding and providing a clearer exposition of the model's reasoning process could enhance the explainability of the evaluation results. However, we would like to clarify our **design considerations**:
>
> 1. While integrating visual grounding and open-form reasoning processes can improve interpretability, it also substantially increases the flexibility of model responses. This **inherent answer diversity** introduces significant challenges for accurate and efficient evaluation, as it is **difficult** to define a unique, objectively correct answer for each question.
> 2. To assess the intermediate reasoning steps of models, the most widely adopted approach is to employ **LLM-as-a-Judge** paradigms. However, compared to multiple-choice questions, such methods can introduce **additional evaluation biases**. As a result, most current benchmarks for Multimodal Large Language Models (such as Video-MME [1], MMMU [2], and MLVU [3]) still predominantly use **multiple-choice formats** to ensure **fair and consistent assessment**.
> 3. We appreciate your suggestion and fully recognize the value of enhancing interpretability, such as by evaluating intermediate reasoning processes. To address this, we plan to **enrich our benchmark** by **incorporating intermediate-step questions**. Specifically, we will design tasks where both the reasoning steps (e.g., visual grounding or intermediate observations) and the final answer are posed as multiple-choice questions. A model will only be considered to have completed the task correctly if it answers both the intermediate and final questions accurately.
> We provide an example below for reference.
>
> **Example**:
>
> - Intermediate Question 1: How many types of product labels are visible on the supermarket shelf in the video?
> - Intermediate Question 2: On which shelf row (from top to bottom) is the most expensive item located?
> - Final Question: What is the unit price of the most expensive item in the video?
>
> We believe this enhancement directly addresses the core of your suggestion, improving our benchmark's ability to evaluate models in a robust and scalable manner. We will incorporate this new evaluation method into our work.
>
> Thank you once again for your valuable feedback and for recognizing our work!
>
>
> [1] Video-MME: The First-Ever Comprehensive Evaluation Benchmark of Multi-modal LLMs in Video Analysis
>
> [2] MMMU: A Massive Multi-discipline Multimodal Understanding and Reasoning Benchmark for Expert AGI
>
> [3] MLVU: Benchmarking Multi-task Long Video Understanding

---

> > ### Comment · Reviewer_kcWJ · 2025-08-06
> >
> > Thank you for your response. My concerns have been addressed so that I will maintain the original score. I have increased my confidence to 4.

---

> > > ### Author Response · Authors · 2025-08-06
> > >
> > > Thank you so much for your reply and recognition of our work!

---

### Official Review · Reviewer_JVrA · 2025-06-23

**Rating:** 4
**Confidence:** 5

**Summary:**

The paper introduces MME-VideoOCR, a novel benchmark designed to comprehensively evaluate the OCR capabilities of Multimodal Large Language Models (MLLMs) in video scenarios. The benchmark includes 10 task categories (25 specific tasks) with 1,464 videos and 2,000 manually annotated QA pairs, rigorously testing 15 state-of-the-art MLLMs. Results show that even the top-performing model (Gemini-2.5 Pro) scores only 1,428 out of 2,000, highlighting the challenge of video OCR. The paper also identifies key weaknesses in current models, particularly in spatio-temporal reasoning and cross-frame semantic understanding.

**Additional Feedback:**

None

**Dataset Code Accessibility:**

Yes

**Dataset Code Comments:**

None

**Ethical Considerations:**

No, there are no or only very minor ethics concerns

**Final Justification:**

The author has addressed most of my concerns, and I am inclined to maintain my positive score.

**Limitations Weaknesses:**

**Weaknesses**：

1. While the dataset is large (1,464 videos), certain niche tasks (e.g., AIGC Videos) may have insufficient samples, potentially affecting evaluation stability.

2. Some tasks are too easy for leading models (e.g., Subtitle-Based Video Understanding), suggesting a need for more challenging samples.

3. The paper notes that higher resolution and frame counts improve performance, but does not analyze the practical feasibility of deploying such models.

4. The dataset primarily focuses on horizontally oriented videos; the impact of vertical videos (common in UGC) is not deeply explored.


**Questions for Authors:**

1.  Does the dataset include enough non-English text to robustly evaluate multilingual OCR performance?

2.  Frame Understanding – The models perform poorly on tasks like Scrambled Recognition. Are there specific architectural changes or training strategies that could help?

3.How does the benchmark account for fast-moving or distorted text (e.g., motion blur, perspective changes)?

4. The adversarial videos use black frames, but are other attack types (e.g., noise injection, occlusion) considered?

**Strengths Contributions:**

1.VideoOCR is the first comprehensive benchmark for video OCR, addressing a critical gap in evaluating MLLMs in dynamic video environments.

2. The dataset spans 44 scenarios, including multilingual, perceptual, comprehension, and reasoning tasks, ensuring broad applicability.

3.  High-quality manual annotation and expert verification minimize biases and errors, enhancing reliability.

4. The benchmark effectively differentiates model performance, revealing clear gaps in capabilities (e.g., temporal integration).

5. The paper provides fine-grained insights into model limitations (e.g., poor performance in Trajectory Recognition and Scrambled Recognition).

---

> ### Author Rebuttal · Authors · 2025-07-30
>
> Thank you very much for your review and valuable comments. We also sincerely appreciate your recognition of our work.
>
> Next, we would like to address your concerns point by point.
>
> **Response to Weakness 1:**
>
> - At the time of data collection, existing AIGC systems exhibited significant limitations in generating text within videos. Even **Wan2.1**, despite its advanced capabilities, often struggled with generating diverse and high-quality text, especially under complex conditions. The generated text tended to be **stylistically uniform**, and models were unable to reliably respond to prompts requiring complex or varied text forms. Thus, given the technical landscape at that time, further increasing the quantity of AIGC videos would **not** have substantially improved the diversity or stability of the evaluation.
> - Based on these constraints, we generated over **1,000** AIGC videos using Wan2.1 and carefully curated **50** high-quality and representative samples through **manual selection**. These videos exhibit substantial diversity in terms of content, style, text appearance, texture, special effects, and timing of text occurrence and disappearance.
> - During sample selection, we observed that AIGC systems frequently produce videos with **missing or erroneous text**, such as “through” rendered as “throuh”. Models tend to misidentify these text errors based on **language priors**. To address this, we deliberately include such challenging cases in our benchmark, further strengthening its coverage and reliability.
> - We carefully considered the allocation of samples across different tasks. Tasks such as Visual Text QA, which reflect **core model capabilities** and exhibit greater diversity in task forms, were assigned larger sample sizes. In contrast, specialized categories such as AIGC and adversarial videos are **less common** in practical scenarios, so their sample sizes are relatively smaller.
> - We fully agree that increasing the number of samples can further enhance the stability of model evaluation for specific tasks. Expanding and improving the benchmark is one of **our main priorities moving forward**. Specifically, we plan to increase the total number of samples from 2,000 to **5,000** and the number of videos from 1,464 to **3,000**, with at least **100** samples for **each specific task category**. For **AIGC videos**, we will include **200** samples, carefully curated to ensure both quality and diversity. To this end, we have already generated **3,000** higher-quality AIGC videos using the more advanced Wan 2.2 and Veo3.
>
> **Response to Weakness 2:**
>
> - Subtitle-Based Video Understanding serves as a **fundamental** task reflecting the core OCR capabilities of models. Despite its relative simplicity, this task is widely applicable in real-world scenarios and remains essential for evaluating basic model competence.
> - While the task may appear straightforward, it is worth noting that many leading models (such as **NVILA 8B and LLaVA-OneVision 7B**) still achieve less than **80%** accuracy. This indicates that even on these basic tasks, there is considerable room for improvement, and the results provide meaningful insights into model performance.
> - Our benchmark also includes a range of more **challenging tasks** specifically designed for advanced models. For example, tasks like Multi-Hop Needle in a Haystack make models achieve less than **30%** accuracy, and for Trajectory Recognition, the accuracy drops to **0%**. Together, these high-difficulty tasks, combined with easier ones like Subtitle-Based Video Understanding, ensure our benchmark offers **a comprehensive range of difficulty levels and strong discriminatory power**.
> - We fully agree that **introducing more challenging samples** would further enhance the evaluation. In future versions, we plan to expand this aspect. For example, in the case of Subtitle-Based Video Understanding, we are collecting and incorporating videos with **multi-person dialogues** (involving multiple speakers’ lines), which significantly increases the difficulty for models to associate textual content with specific video segments.
>
> **Response to Weakness 3:**
>
> - Increasing input resolution and frame count requires processing a significantly larger number of tokens. This essentially tests the model’s **ability to handle long-sequence inputs**. In real-world deployment, however, the context window limitation of current models often makes it infeasible to process such large token volumes. As a result, practical deployments typically resort to strategies such as sparse frame sampling or reducing input resolution as a **trade-off**.
> - Recently, several works have focused on **extending context window length**, such as InternVideo2.5 [1], Eagle 2.5 [2], and VideoChat-Flash [3]. These models are capable of efficiently handling longer token sequences, making it increasingly feasible to deploy models that utilize higher resolutions and frame counts in practical settings.
> - We will include a detailed discussion and analysis of these technical solutions and their respective advantages and limitations in the new version of our paper, providing a more comprehensive reference for the feasibility of deploying such models in real-world scenarios.
>
> **Response to Weakness 4:**
>
> - Vertical videos are an important component in our benchmark and play a **key role** in achieving comprehensive evaluation. First, vertical video is a common format for content captured on mobile devices and is prevalent on platforms such as TikTok and other short video services. The proportion of vertical videos in online video data is steadily increasing, making effective OCR-based understanding of this format a crucial direction for both future model development and real-world applications. Second, since most current models are trained primarily on horizontally oriented videos, evaluating their performance on vertical videos provides a strong test of the model’s generalization capability and practical effectiveness.
> - Our dataset specifically includes **137** vertical videos and **182** corresponding QA pairs, accounting for **9.4%** and **9.1%** of the total, respectively, reflecting our careful consideration of this format. In the next version, we plan to further expand the number of vertical video samples to increase their proportion to **20%**, thus enhancing the comprehensiveness of the benchmark and aligning with current trends in video content.
> - Due to space limitations, we did not elaborate on this aspect in the paper. We appreciate your attention to this point and will include a detailed discussion in the revised version.
>
> **Response to Question 1:**
>
> - Our dataset consists of 1,464 videos and 2,000 QA pairs, including **153** Chinese videos and **212** corresponding Chinese QA pairs, representing **10.5%** and **10.6%** of the total, respectively. This substantial number of samples ensures robust evaluation of models’ bilingual OCR capabilities.
> - We are currently collecting additional videos and plan to expand the number of Chinese videos to **1,000** (accounting for **33%**) in the new version of our benchmark, along with **2,000** corresponding QA pairs (accounting for **40%**), in order to provide a more comprehensive evaluation.
>
> **Response to Question 2:**
>
> - Most current video understanding models are trained primarily on **sequentially played videos**. As a result, these models have limited exposure to data with shuffled or non-sequential playback patterns during training.
> - This uncommon playback mode requires models to effectively **align spatial positions and integrate temporal information** across multiple frames. Such complex compositional reasoning has not been the focus of previous training regimes, leading to a lack of robust capabilities in handling these tasks.
> - To address this limitation, there are two main directions for improvement: (i) deliberately constructing video datasets that feature shuffled playback or key information distributed out of order, and (ii) emphasizing the training of models’ spatio-temporal grounding and long-sequence information integration abilities.
>
> **Response to Question 3:**
>
> - We conducted **careful manual screening** during dataset construction to ensure that the selected videos comprehensively cover these special scenarios. Additionally, we designed corresponding questions to specifically target and assess model performance on frames containing motion blur, perspective distortions, and other challenging text appearances. This process ensures that our benchmark reliably evaluates models’ robustness in handling such real-world complexities.
>
> **Response to Question 4:**
>
> - We have experimented with using noisy videos as adversarial samples and evaluated their impact on several models, including LLaVA-OneVision 7B, LLaVA-Video 7B, and Qwen2.5-VL 32B. The results showed that the accuracy on noisy videos was very similar to that on black-frame videos. For consistency across evaluations, we therefore opted to use black-frame videos as the standard adversarial attack type.
> - The **primary motivation** for introducing adversarial videos is to **simulate real-world disruptions** such as screen flicker or information loss, which are common in practical scenarios. In this regard, both black frames and noise injection serve a similar purpose by obscuring information. As for partial occlusion of text, this is a frequent phenomenon in real-world videos (e.g., tree branches partially covering a billboard) and, based on annotator feedback, such cases are already well represented across other task categories in our dataset. Defining occlusion as a separate adversarial category would result in redundant sample distributions.
>
> [1] InternVideo2.5: Empowering Video MLLMs with Long and Rich Context Modeling
>
> [2] Eagle 2.5: Boosting Long-Context Post-Training for Frontier Vision-Language Models
>
> [3] VideoChat-Flash: Hierarchical Compression for Long-Context Video Modeling

---

> > ### Comment · Reviewer_JVrA · 2025-08-04
> >
> > The author has addressed most of my concerns, and I am inclined to maintain my positive score.

---

> > > ### Author Response · Authors · 2025-08-04
> > >
> > > Thank you very much for your constructive feedback!
> > >
> > > We will improve our work based on your suggestions and make the corresponding revisions in the new version.
> > >
> > > Once again, we sincerely appreciate your recognition of our work!

---

> > ### Author Response · Authors · 2025-08-04
> > **Supplementary experiments to address Question 4**
> >
> > Dear Reviewer:
> >
> > To better address **Question 4**, we conduct a set of supplementary experiments.
> >
> > Specifically, we evaluated a diverse range of models including **LLaVA-OneVision 7B**, **LLaVA-Video 7B**, **Oryx-1.5 7B**, **Qwen2.5-VL 7B**, **Oryx-1.5 32B**, **Qwen2.5-VL 32B**, and **Qwen2.5-VL 72B**. These models vary in both frame **sampling strategies** and **model scales**, making them suitable for a **comprehensive** evaluation of the impact of **different adversarial attacks** (e.g., black frame, white frame, diverse noise injection).
> >
> > The experimental results (accuracy) are summarized as follows:
> >
> > | Model               | Black Frame | White Frame | Gaussian Noise | Poisson Noise | Uniform Noise | Maximum Difference |
> > |--------------------|-------------|-------------|----------------|----------------|----------------|---------------------|
> > | LLaVA-OneVision 7B | 66.0%       | 66.0%       | 64.0%          | 66.0%          | 66.0%          | 2.0%                |
> > | LLaVA-Video 7B     | 66.0%       | 64.0%       | 64.0%          | 68.0%          | 66.0%          | 4.0%                |
> > | Oryx-1.5 7B        | 72.0%       | 72.0%       | 72.0%          | 72.0%          | 68.0%          | 4.0%                |
> > | Qwen2.5-VL 7B      | 80.0%       | 80.0%       | 78.0%          | 78.0%          | 80.0%          | 2.0%                |
> > | Oryx-1.5 32B       | 72.0%       | 76.0%       | 72.0%          | 72.0%          | 74.0%          | 4.0%                |
> > | Qwen2.5-VL 32B     | 84.0%       | 84.0%       | 82.0%          | 82.0%          | 84.0%          | 2.0%                |
> > | Qwen2.5-VL 72B     | 90.0%       | 86.0%       | 88.0%          | 88.0%          | 86.0%          | 4.0%                |
> >
> > These attacks exhibit **different** characteristics—for example, the various noise types **simulate** perturbations of different intensities and distributions. This provides a foundation for comprehensively evaluating the impact of different attack types.
> >
> > From the experimental results, we observe that the maximum accuracy difference caused by these attacks is only **4.0%**. This relatively small variance suggests that the choice of attack type has a **limited** impact on model performance. Instead, the **key factor** remains whether the model can effectively process **more** visual frames and perform reasonable **inference** and **completion** based on **contextual information**, even when part of the input is perturbed.
> >
> > We hope that these additional experiments help to further address your concern and improve the completeness of our work.
> >
> > We also sincerely look forward to your further recognition and support!
> >
> > Best wishes!

---

### Official Review · Reviewer_Wpk6 · 2025-07-01

**Rating:** 5
**Confidence:** 4

**Summary:**

This manuscript propose to evaluate MMLM on OCR related capability under video context. Namely, the proposed benchmark MME-VideoOCR composes 10 task categories, 25 individual tasks and spans 44 diverse scenarios. They also evaluated 15 state-of-the-art MLLMs on MME-VideoOCR and show the headroom of improvement is large (1500 out of 2000).

**Dataset Code Accessibility:**

Yes

**Ethical Considerations:**

No, there are no or only very minor ethics concerns

**Final Justification:**

I have read the rebuttal and confirm my final rating as it.

**Limitations Weaknesses:**

- In Table 4 and Table 5, it would be helpful to add a column/row to indicate how many examples for each task.
- What are the performance of some other proprietary LLMs including ChatGPT and Claude?
- The paper states "in the MME-VideoOCR evaluation, the number of input frames was uniformly set to 64 for all models", I wonder how to handle the videos with less than 64 frames.
- In task categories,  "Long Videos", "Multi-Hop Needle in A Haystack" and "Temporal Grounding" are the ones that models performs exceptionally bad. Do authors have insights why?
- As a pressure test to Fig.4, at which resolution and number of frames would the model performance on the benchmark saturate?

**Strengths Contributions:**

The paper is written in clear language and very easy to follow. The angle of video OCR is indeed valuable in video benchmarks.

---

> ### Author Rebuttal · Authors · 2025-07-30
>
> We sincerely thank you for the time and effort you devoted to reviewing our paper, and we truly appreciate your recognition of our work.
>
> We have provided detailed responses to each of your concerns and hope they adequately address the issues you raised.
>
> **Response to Weakness 1:**
>
> Thank you for your helpful suggestion. We agree that **adding** a column or row to indicate the number of examples for each task would make the sample distribution across different tasks clearer. We will revise **Tables 3, 4, 8, 9** in the new version to include this information.
>
> **Response to Weakness 2:**
>
> - We have tested **GPT-4o-0806**, and its performance ranks just below Gemini-2.5, Qwen2.5-VL 72B, and InternVL3 78B. The specific accuracy scores are as follows:
> | Model   | Size | TR    | VTQA  | TG    | AR    | CDT   | STP   | CFTU  | TBR   | TBVU  | RVT   | Total |
> |---------|------|-------|-------|-------|-------|-------|-------|-------|-------|-------|-------|--------|
> | **Samples**  |  -   | 300 | 300 | 200 | 150 | 200 | 250 | 150 | 150 | 200 | 150 | 2000  |
> | GPT-4o-0806  |  -   | 83.3% | 81.6% | 60.5% | 74.7% | 51.5% | 68.0% | 30.7% | 60.7% | 59.0% | 75.3% | 66.4%  |
> - Due to resource limitations, we have not yet been able to obtain access to the Claude API. We are actively seeking access and will include results for Claude as soon as possible.
>
> **Response to Weakness 3:**
>
> - If a video contains fewer than 64 frames, we use all available frames as input. For videos with more than 64 frames, we sample 64 frames according to the default frame sampling strategy of the model.
> - This can be **summarized** as: $ \text{Input Frame Num} = \min(64, \text{actual number of frames}) $
>
> **Response to Weakness 4:**
>
> - **Long Videos.** Long videos contain many frames and a large volume of information. Due to **context window limitations**, most models (e.g., LLaVA-Video, LLaVA-OneVision) can only input a subset of key frames using **sparse sampling** strategies. This inevitably leads to substantial information loss. Although some models (such as Qwen2.5-VL, VideoChat-Flash, and VideoLLaMA 3) attempt to increase the number of input frames by **reducing resolution** or applying **compression** techniques, these methods often **degrade** the structural integrity of text information, making it difficult for the model to accurately process and recognize text.
> - **Multi-Hop Needle in a Haystack.** This task requires the model to frequently jump between frames based on textual clues and accurately recognize and understand the visual content across these hops. Such complex reasoning demands strong **memory and long-sequence reasoning capabilities**. Additionally, from a **data perspective**, current video understanding models are mostly trained on sequential video datasets and rarely exposed to scenarios with shuffled playback or non-chronological distributions of key information. This training gap further limits their performance on such tasks.
> - **Temporal Grounding.** Videos often contain significant redundant information, making it challenging for models to **focus** on the segments most relevant to the question. Moreover, most current models **lack** dedicated temporal modeling modules and typically rely on simply **concatenating** multiple frame inputs. Some recent approaches have introduced improvements, such as VideoChat-Flash with explicit timestamp coordinates and Qwen2.5-VL with M-RoPE position encoding. However, these solutions still fall short for precise temporal grounding, indicating a need for further research and optimization.
>
> **Response to Weakness 5:**
>
> - **Impact of Input Frame Number.** We conducted additional experiments with **Qwen2.5-VL-7B, InternVL3-8B, LLaVA-Video-7B, and Oryx-1.5-7B**. We found that model performance tends to saturate when the number of input frames reaches **64**. Increasing the input further to **80** or **96** frames did not yield significant performance gains and, in some cases, even led to slight decreases. We attribute this to the difficulty models face in retrieving relevant textual information from excessive **redundant** frames, which can introduce noise and hamper overall accuracy.
> - **Impact of Input Resolution.** We also evaluated the impact of input resolution using **Qwen2.5-VL-7B, InternVL3-8B, and Oryx-1.5-7B**. Our results indicate that different models reach performance saturation at **different** resolution settings. This variation may be due to differences in high-resolution image processing strategies (e.g., Qwen2.5-VL leverages M-RoPE) as well as the quality and diversity of training data (for example, Qwen2.5-VL benefits from high-quality, high-resolution OCR training samples).
>
> Once again, thank you for your valuable feedback and suggestions. We sincerely hope our responses address your concerns.

---

> > ### Comment · Reviewer_Wpk6 · 2025-08-06
> > **Re w5 response**
> >
> > I would appreciate the author to provide more detailed data points to support the # frames and various resolution results to complement the benchmark analysis (what are the saturation points and where is the performance limit by naively increasing computing budget). These information included in the main paper or appendix is valuable for other researchers to understand both the benchmark and models' limit on OCR task in video domain.

---

> > ### Author Response · Authors · 2025-08-07
> > **Detailed Data**
> >
> > Thank you very much for your professional and insightful feedback, which provides valuable guidance for improving our work.
> >
> > Below, we present detailed analysis and experimental results to address your concern regarding the impact of the number of input frames and input resolution on benchmark performance, including saturation points and performance limits.
> >
> > - **Impact of Input Frame Number**
> >
> >   We evaluate 4 representative models—Qwen2.5-VL-7B, LLaVA-Video-7B, InternVL3-8B, and Oryx-1.5-7B—under varying **maximum input frame numbers**: 8, 16, 32, 64, 80, and 96. The results are shown below:
> >
> >   | Model             | 8F   | 16F   | 32F   | 64F   | 80F   | 96F   |
> >   |-------------------|------|-------|-------|-------|-------|-------|
> >   | Qwen2.5-VL-7B     | 55.3 | 58.2  | 59.5  | 59.1  | 58.75 | 58.3  |
> >   | LLaVA-Video-7B    | 48   | 50.65 | 51.75 | 52.8  |   -   |   -   |
> >   | InternVL3-8B      | 57.15| 59.1  | 59.95 | 59.8  |   -   |   -   |
> >   | oryx-1.5-7B       | 44.4 | 47.85 | 48.7  | 49.6  | 48.9  | 48.55 |
> >
> >   - We observe that, with fixed resolution, performance generally **improves** with more input frames, up to around **64** frames where most models **saturate**.
> >
> >   - Due to architectural constraints such as **context window size**, some models (e.g., LLaVA-Video-7B, InternVL3-8B) are unable to process more frames.
> >
> >   - Interestingly, for Qwen2.5-VL-7B and oryx-1.5-7B, **increasing** the number of frames beyond **64** (to 80/96) leads to obvious performance **degradation**. We hypothesize that this is due to the model’s limited capacity in handling **long sequences**, which may hinder its ability to **focus** on key information—potentially due to limitations in **attention allocation** or **memory compression** strategies.
> >   - This finding suggests that increasing the number of input frames alone is **not** sufficient; robust long-sequence modeling is also **essential**.
> >
> > - **Impact of Input Resolution**
> >
> >   - We further evaluate model performance by scaling the **longer edge** of input frames to: 224, 336, 448, 560, 672, 784, and 896 pixels, while fixing the maximum number of frames to 32. Results are as follows:
> >   | Model            | 224  | 336   | 448   | 560   | 672   | 784   | 896   |
> >   |------------------|------|-------|-------|-------|-------|-------|-------|
> >   | Qwen2.5-VL-7B    | 40.4 | 45.55 | 49.75 | 52.95 | 55.15 | 57.3  | 59.15 |
> >   | VideoLLaMA 3     | 35.9 | 43.95 | 48.65 | 51.75 | 51.6  | 51.25 | 51.1  |
> >   | oryx-1.5-7B      | 35.8 | 37.95 | 42.9  | 46    | 48.2  | 49.4  | 49.1  |
> >
> >   - These results show a consistent performance **gain** with higher resolution, followed by saturation. However, the **saturation** points **differ** significantly across models: 896 for Qwen2.5-VL-7B, 672 for VideoLLaMA 3, and 784 for oryx-1.5-7B.
> >   - We attribute these differences to two **main factors**:
> > 	1.	Model design and compression strategies. For instance, Qwen2.5-VL employs M-RoPE, while Oryx-1.5 adopts a dynamic compression strategy with higher compression ratios.
> > 	2.	Training data quality and diversity. Qwen2.5-VL benefits from large-scale, high-resolution OCR training data, contributing to better high-resolution performance.
> >
> >   - Thus, the resolution-performance limit is **model-specific** and depends on both architectural design and data-driven learning capacity.
> >
> > We will include the detailed data in the new version and provide a comprehensive analysis.
> >
> > Once again, we appreciate your helpful suggestion, and we sincerely hope that the additional analysis and results will address your concern.

---

> > > ### Comment · Reviewer_Wpk6 · 2025-08-07
> > >
> > > Thank you for your reply. I acknowledge the contribution of this work and think it well complement to current video understanding research.

---

> > > > ### Author Response · Authors · 2025-08-08
> > > >
> > > > Thank you so much for your recognition of our work! We will revise the corresponding part in the new version.

---

### Official Review · Reviewer_foB1 · 2025-07-19

**Ethics Flags:** Data privacy, copyright, and consent
**Rating:** 5
**Confidence:** 3

**Summary:**

This paper addresses the important and challenging task of Optical Character Recognition (OCR) in videos by introducing a comprehensive benchmark, MME-VideoOCR, to evaluate how state-of-the-art video MLLMs perform in different scenarios. The authors identify 10 common video scenarios where OCR plays a critical role and design 25 evaluation tasks accordingly. They evaluate 15 state-of-the-art MLLMs, analyzing performance across model size and frame sampling strategies.

**Dataset Code Accessibility:**

Yes

**Ethical Comments:**

There is an ethical concern regarding the use of public video data. The authors should clearly state:

1. Whether they obtained permission or licenses (from the creator) to use the public videos (e.g., YouTube) as the secondary usage, especially for AI benchmarking.

2. How privacy concerns were handled, particularly for videos containing identifiable human subjects.

**Ethical Considerations:**

Yes, there are significant ethics concerns that require review by an ethics expert

**Final Justification:**

The authors addressed most of my concerns especially the privacy ones. Good work so I raised my score.

**Limitations Weaknesses:**

1. Unclear annotation and screening process: The details of the human annotation process are insufficient. No annotation guidelines, annotator background, or composition are provided, making it unclear how consistent the process is. For example, what standard does the second-stage expert follow to select “1–2 high-quality QA pairs” from the initial “3–4 QA pairs”? Who are these “experts”? If only a single expert reviews each data sample, potential selection bias exists. If multiple reviewers are involved, inter-annotator agreement should be reported.

2. Concerns about MLLM-as-the-judge in video selection: The use of GPT-4o as the only judge for selecting videos can also cause selection bias. The judging criteria (e.g., what qualifies as “sufficient visual dynamics” or “semantically meaningful text”) are not clearly defined, leaving the video sampling process less transparent.

3. **Ethical concerns regarding copyright and privacy**: A major concern is the use of public videos (e.g., YouTube) without clear statements on copyright, licensing, or permissions for AI deployment. Additionally, videos containing human subjects may raise privacy concerns. The authors should explicitly state whether any licenses,  IRB approvals, or ethical review procedures were followed.

Note: My score could change if the rebuttal clearly addresses the ethical concerns.

**Strengths Contributions:**

1. Well-motivated and comprehensive benchmark design: The benchmark is logically designed, with a clear motivation rooted in the importance of video OCR. The 10 scenarios are thoughtfully categorized based on OCR-related capacities, and the resulting 25 evaluation tasks provide broad coverage of relevant challenges for video MLLMs.

2. Novel and high-quality dataset: The dataset construction is commendable. The authors annotate new videos from public platforms and regenerate QA pairs even for videos sampled from existing benchmarks. The two-stage human screening and annotation process improves data quality, contributing a valuable resource for the community.

3. Comprehensive evaluation and insightful analysis:
The evaluation covers a diverse set of 15 MLLMs, including open- and closed-source models, efficient models, and long-video models. The analysis is conducted from meaningful perspectives of model size and frame sampling strategy.  The findings are also insightful. For instance, the paper highlights that current MLLMs still struggle with real long-context reasoning, even though they perform adequately on information retrieval or short-term temporal understanding.

Overall, this work can be viewed as a good/standard benchmark and dataset paper, making it well-suited for this track.

---

> ### Author Rebuttal · Authors · 2025-07-31
>
> Thank you very much for your valuable and thoughtful comments, which have provided constructive guidance for improving our work.
>
> We would like to address your concerns point by point.
>
> **Response to Weakness 1:**
>
> - **Annotator Background**: All annotations were conducted by a professional team of **20** annotators with substantial prior **experience** in AI-related data annotation, especially in the video domain. All team members hold at least a **bachelor’s degree** and possess strong proficiency in both **English and Chinese**. They have undergone **training** specific to this project, including detailed instructions to ensure quality and consistency.
> - **Expert Background**: The second-stage experts are **co-authors** of this paper, all of whom are **PhD students or researchers** in Computer Science/AI. They have significant expertise in Multimodal Large Language Models and are actively engaged in related research.
> - **Screening Protocol**: In the second-stage quality check, each data sample—consisting of 3–4 candidate QA pairs—was independently reviewed by **2** experts. These experts followed a standardized guideline (partially shown below) to assess informativeness, reasoning depth, and answer accuracy. The **top** 1–2 QA pairs were selected only if both experts independently **agreed** on their quality. In case of disagreement, the sample was either revised or sent back for re-annotation to avoid selection bias and maintain high consistency.
> - **Annotation and Selection** **Guidelines**: To address your concern about missing documentation, we would like to provide a simplified version of our annotation and screening guideline below. Due to space limitations in the rebuttal, the full guidelines will be included in the supplementary material of the revised submission.
> - **Annotation Guidelines**
>   - **Video Quality Standards**: To qualify for annotation, each video must meet the following criteria:
>      - **Sufficient Motion Dynamics**: The video should contain noticeable dynamic elements, such as character movements, scene transitions, background changes, shifts in the main subject, or camera movements.
>      - **Presence of Visible Text**: The video must include clearly legible text elements that can be reliably annotated.
>      - **Content Safety and Privacy**: Videos containing harmful, offensive, or privacy-invading content (e.g., personal identification, sensitive scenes) are strictly prohibited and must be excluded from the dataset.
>   - **Text Quality Standards**: Text instances selected for annotation must adhere to the following principles:
>      - **Semantic Relevance**: Text must carry meaningful semantic content and exhibit a clear contextual relationship with the visual scene, such as product labels, road signs, posters, storefront signs, etc.
>      - **No Random Character Strings**: Random or meaningless sequences of characters should not be included.
>      - **Special Text Forms**: The inclusion of special forms of text such as subtitles and watermarks should be controlled and kept within specified proportions as required by the task.
>   - **QA Design Standards**: For each video, QA pairs should be constructed according to the following guidelines:
>      - **Task-Oriented Design**: Based on the task type (e.g., Text Recognition, Visual Text QA), each video should have 3 to 4 QA pairs that reflect the target capabilities of the model.
>      - **Answer Visibility and Clarity**: Each question must correspond to an answer that is clearly visible in the video and free of ambiguity.
>      - **Video-Dependent Questions**: Avoid designing questions that can be answered without watching the video (e.g., “In what year was the United States founded?”). Such questions lead to answer leakage and violate the intended purpose of visual text grounding.
>
> **Response to Weakness 2:**
>
> - **Hybrid Selection Strategy to Ensure Diversity and Minimize Bias.** We adopted a hybrid pipeline that combines automated filtering using GPT-4o with human verification. GPT-4o was used to efficiently screen a large pool of candidate videos based on prompts we carefully designed to capture the required characteristics (see below). This automated stage helps ensure source diversity across domains and content types. Following this, human annotators reviewed the selected videos during the QA annotation phase to further verify the relevance, diversity, and safety of the content. This two-stage process balances efficiency and quality, and helps mitigate potential selection bias introduced by relying solely on either automated or manual selection.
> - **Defined Judging Criteria via Prompt Design.** To make the filtering process more transparent, we explicitly incorporate evaluation criteria in the GPT-4o prompt along two main axes:
>     - **Visual dynamics**: The video must exhibit clear visual changes, such as character actions, scene transitions, background shifts, object movements, or the appearance/disappearance of visual elements (including text).
>     - **Semantic text presence**: The video must contain text that is meaningful and contextually related to the visual content. We explicitly instructed the model to avoid selecting videos with irrelevant or meaningless text sequences, and to ensure the text is not dominated by subtitles, watermarks, or logos. A controlled proportion of these special text forms was allowed to reflect real-world scenarios while maintaining relevance.
>
> **Response to Weakness 3:**
>
> - **Video Sources**: Our dataset is composed of **2** parts: (1) videos collected from existing academic datasets, and (2) publicly available videos sourced from online platforms.
> - **Use of Existing Datasets**: For videos sourced from prior academic datasets (e.g., BOVText, EgoTextVQA), we carefully reviewed their **original licensing** terms and usage policies. These datasets were released for research purposes, and their usage is **permitted** under their respective open-source licenses. In addition, we contacted the authors of key datasets via email and obtained explicit **permission** to use their data for academic benchmarking.
> - **Public Video Collection**: When sourcing videos from public platforms, we prioritized those that do **not** feature identifiable individuals or private scenarios. In cases where **faces** were present, we applied automatic **face detection** and **blurring** techniques to protect personal identity and comply with privacy standards. All selected videos **exclude** sensitive or private content.
> - **Licensing, Usage Declaration, and Contact Mechanism**: All collected videos are used strictly for **academic research** purposes. We do **not** claim ownership over any of the video content. We clearly state this in our dataset documentation on GitHub and HuggingFace, where disclaimers are prominently **displayed**. We also provide a public **contact email** for video owners to reach us; we are committed to promptly addressing any takedown or modification requests. Further clarification on our data usage policy will be **added** in the revised version of the paper.
>
> Thank you once again for raising important concerns regarding **privacy and ethics**. We deeply value these issues, and your feedback is highly constructive in helping us further **improve** our work.
>
> We have made every effort to comply with **open-source licenses and privacy regulations**. Specifically, we carefully **avoided** collecting videos that involve private scenes or sensitive personal information. Additionally, facial regions in the videos have been processed with **blurring** techniques to mitigate potential privacy risks. We have also made efforts to ** contact**  the original video owners and dataset creators for all collected videos whenever possible.
>
> Our goal is to strike a careful **balance**  between advancing academic research and upholding strong ethical and privacy standards.
>
> We would like to reiterate that we **strictly**  adhere to academic open-source protocols and relevant copyright and privacy norms. All rights to the videos remain with the original content **owners** , and we are committed to responding promptly and appropriately to any related concerns.

---

> ### Author Response · Authors · 2025-08-04
> **Follow-up on rebuttal – kindly requesting reviewers’ feedback**
>
> We would like to kindly follow up on our previous rebuttal, especially regarding the important concern raised about **dataset ownership**, **privacy**, and **ethical considerations**.
>
> We want to reaffirm that privacy and ethical compliance were carefully addressed during the dataset collection and curation process:
> - The majority of our video samples are derived from existing datasets, re-annotated under their respective **open-source licenses**. We also actively contacted the original dataset authors (e.g., **EgoTextVQA**, **BOVTex**t) where possible, and received their **consent** for reuse.
> - For AIGC (synthetic) content, we explicitly indicate the nature of each video and **disclose** the generative model used.
> - For web-crawled public content, we apply **strict filtering** to exclude any material that might involve personal privacy or sensitive locations.
> - For videos involving human subjects, we conduct **face detection** and **blurring**, followed by **manual review** to ensure privacy protection.
> - We clearly **state** in the **GitHub** and **HuggingFace** that we comply with the relevant **CC BY 4.0 license** (same as **OpenVid-1M** [1] and **VideoUFO** [2], we have carefully checked that we follow this license), that the dataset is intended for **academic use only**, and that **all rights** remain with the **original video owners**. We are committed to actively responding to and addressing any concerns related to copyright or privacy (we provide the **contact e-mail** in all public releases.).
>
> We are committed to promptly addressing any concerns raised by the community and welcome further feedback. **Our goal** is to advance research while fully respecting ethical and legal standards.
>
> We sincerely hope you could reconsider our clarifications in the rebuttal, and we would greatly appreciate your response.
>
> [1] OpenVid-1M: A Large-Scale High-Quality Dataset for Text-to-video Generation
>
> [2] VideoUFO: A Million-Scale User-Focused Dataset for Text-to-Video Generation

---

> > ### Comment · Reviewer_foB1 · 2025-08-04
> > **Thanks and raised my score.**
> >
> > Dear authors,
> >
> > Thank you for your rebuttal, which addressed most of my concerns, especially the privacy part. Given the quality and contribution of this work, I will raise my score. Good luck.

---

> > > ### Author Response · Authors · 2025-08-04
> > >
> > > Thank you very much for your response and for your recognition of our work!
> > >
> > > We truly appreciate your constructive feedback, which is so valuable for improving our paper. We will revise the corresponding parts in the new version.

---

### Note · Authors · 2025-08-12

We are grateful for the effort all reviewers and ACs have dedicated to reviewing our paper. We would like to briefly summarize the reviewers’ feedback and our subsequent efforts during the rebuttal stage.

- **Reviewers’ Appreciate**: The reviewers expressed high appreciation for the novelty, significance, and completeness of our work:
  - Reviewer foB1: "This paper addresses the **important** and challenging task"; "Overall, this work can be viewed as **a good/standard benchmark** and dataset paper, making it well-suited for this track."
  - Reviewer Wpk6: "The angle of video OCR is **indeed valuable** in video benchmarks. I acknowledge the contribution of this work and think it well complement to current video understanding research."
  - Reviewer JVrA: "VideoOCR is the **first comprehensive benchmark** for video OCR .... The paper provides **fine-grained insights** into model limitations ..."
  - Reviewer kcWJ: "Compared to previous approaches, it includes a broader range of tasks and offers **a thorough evaluation** of MLLM"; "Its support for bilingual content **contributes significantly** to the development of the field."
- **Our Rebuttal Efforts and Improvements**: During the rebuttal phase, we conducted additional experiments and analyses to address the reviewers’ concerns, further strengthening our work:
  - Reviewer foB1
    - **Strict adherence** to open-source and privacy protocols.
    - **Detailed** our dataset construction and manual annotation protocol:
  - Reviewer Wpk6
    - **Added** experiments on the closed-source model GPT-4o-0806.
    - **Conducted** extensive performance saturation analyses.
  - Reviewer JVrA
    - **Introduced** comparative experiments with adversarial sample types.
    - **Analyzed** reasons for underperformance in certain tasks.
  - Reviewer kcWJ
    - **Justified** our choice of multiple-choice evaluation format over LLM-as-a-Judge.
    - **Introducing** intermediate questions to enhance interpretability and robustness.

In summary, our work MME-VideoOCR presents the **first comprehensive** benchmark for **video OCR**, with **bilingual support and rigorous privacy safeguards**.

We believe our contributions address **a critical gap in evaluating MLLMs for dynamic videos**, and we hope you will recognize the value and impact of this work.

---

### Decision · Program_Chairs · 2025-09-18

**Decision:**

Accept (poster)

**Comment:**

The paper introduces a comprehensive benchmark MME-VideoOCR to evaluate models' performance on OCR in videos. The authors engaged thoughtfully with the reviewers and addressed their concerns during the rebuttal. All reviewers are supportive of acceptance (scores of 5, 5, 4, 5). The paper is well-written, with detailed analysis, and may be of broad interest to the NeurIPS D&B audience.

===== FINAL UPDATE FROM DB Track PCs ====

The final decision for this paper has been taken by the program chairs after consultation with the SACs. All Senior Area Chairs have ranked papers according to the feedback from the AC during the review process. We decided to leave the original meta-review to reflect the opinion of the AC in light of the initial discussions with reviewers and SAC.